# UPCORE: Utility-Preserving Coreset Selection for Balanced Unlearning

## Abstract

User specifications or legal frameworks often require information to be removed from pretrained models, including large language models (LLMs). This requires deleting or "forgetting" a set of data points from an already-trained model, which typically degrades its performance on other data points. Thus, a balance must be struck between removing information and keeping the model's other abilities intact, with a failure to balance this trade-off leading to poor deletion or an unusable model. To this end, we propose UPCORE (Utility-Preserving Coreset Selection), a method-agnostic data selection framework for mitigating collateral damage during unlearning. Finding that the model damage is correlated with the variance of the model's representations on the forget set, we selectively prune the forget set to remove outliers, thereby minimizing model degradation after unlearning. Across three standard unlearning methods, UPCORE consistently achieves a superior balance between the competing objectives of deletion efficacy and model preservation. To better evaluate this trade-off, we introduce a new metric, measuring the area-under-the-curve (AUC) across standard metrics. Our results show that UPCORE improves both standard metrics and AUC, benefiting from positive transfer between the coreset and pruned points while reducing negative transfer from the forget set to points outside of it. We include our code in the supplementary.

## 1 Introduction

The widespread deployment of ML models, especially large language models (LLMs), has raised concerns around privacy, regulation, and ethical use. Trained on massive, uncurated web data, these models often memorize sensitive, copyrighted, or undesirable content (Shokri et al., 2017; Carlini et al., 2019). With regulations like the GDPR and CCPA granting individuals the "right to be forgotten," efficient methods for removing specific data or topics from pre-trained models are increasingly necessary. Machine unlearning offers a promising solution by enabling targeted data removal without full retraining and also helps reduce harmful outputs, protect intellectual property, and align LLMs with societal values (Jang et al., 2023). These challenges have driven renewed interest in improving model editing and unlearning techniques (Liu et al., 2024; Hase et al., 2024).

Given the growing use of LLMs, prior work has proposed methods for removing knowledge or skills (Cao & Yang, 2015; Bourtoule et al., 2021; Nguyen et al., 2022) and steering behavior in targeted ways (Sinitsin et al., 2020; Meng et al., 2022). However, such editing often induces unintended side effects, reducing utility on unrelated tasks. Effective unlearning therefore requires balancing deletion of undesired information with preservation of overall model utility. To evaluate this, current methods assess both deletion success (via "forget set" performance) and collateral effects on unrelated behaviors (via "retain set" accuracy). This is particularly important in realistic, topic-level unlearning scenarios, such as removing information about an individual or sensitive domain (Li et al., 2024), where deletion may cause over-generalization and degrade performance on semantically or structurally similar inputs.

A key gap in existing research lies in understanding the specific data characteristics that drive over-generalization and collateral effects during unlearning. While prior work (Sheshadri et al., 2024; Chowdhury et al., 2024) has measured damage resulting from unlearning, it does not investigate how attributes of the data – such as its variance – contribute to collateral damage or whether these attributes can be controlled to optimize the trade-off between deletion efficacy and utility retention.

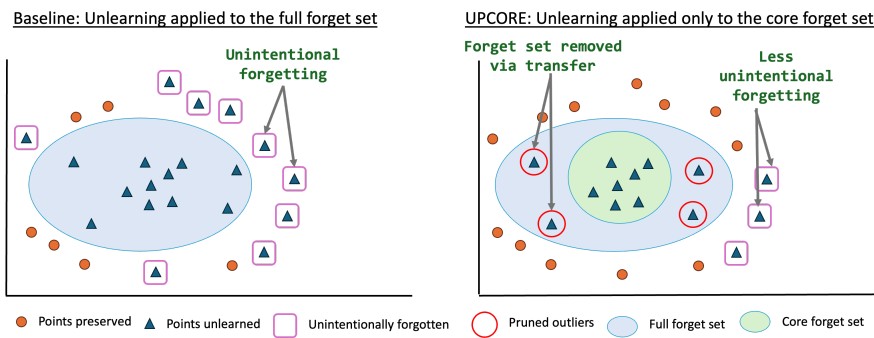

Figure 1: **Left:** Standard unlearning methods are applied equally to all points in the forget set. Here, outlier points in the model's hidden space (visualized in 2D) contribute to the unintentional forgetting of points outside of the forget set (i.e. collateral damage). **Right:** By finding a lower-variance coreset within the forget set, UPCORE reduces damage while maintaining forget performance via positive transfer from the coreset to the pruned points.

Focusing on a topic-based setting where the forget set comprises semantically coherent groups of information, we seek to address these questions:

1. *What measurable attributes of the forget set drive collateral effects during unlearning?*
2. *Can these attributes be systematically controlled to optimize the trade-off between deletion effectiveness and model utility?*

We investigate which properties of the forget data correlate with collateral damage during unlearning. Our analysis reveals a strong positive correlation between the variance of the model's hidden state representations corresponding to datapoints in the forget set (hidden state variance, or HSV), and the extent of collateral damage to the model after unlearning. In other words, unlearning a set of widely-distributed datapoints (as shown in Fig. 1 (left)) leads to more damage than unlearning a more densely-distributed set. Building on this insight, we hypothesize that selectively curating a lower-variance coreset from the larger forget set can help optimize this trade-off, as shown in Fig. 1 (right).

To this end, we introduce UPCORE, which constructs a core forget set by systematically identifying and pruning data points in the forget set that contribute most to the variance and thereby to collateral damage. UPCORE organizes points into an Isolation Forest (Liu et al., 2008a), which identifies anomalous points in a set. By pruning these points, we reduce the variance within the forget set, which we find leads to less damage. Crucially, in addition to reducing collateral damage, UPCORE in fact leverages it by identifying two separate kinds of collateral effects: (1) **Negative collateral damage**: Unintended degradation of unrelated model capabilities and (2) **Positive collateral transfer**: The intended impact on pruned data points removed to form the core forget set. This is illustrated in Fig. 1, where pruned outlier points are still unlearned – despite not being a part of the coreset used for unlearning – due to positive transfer, and is further highlighted by our results in Table 2 and Table 4, which show that UPCORE results in better unlearning than a randomly-selected subset while also having better knowledge retention on non-forget data. We show positive collateral transfer enabled by UPCORE in Table 5, with deletion transferring to points outside the coreset. Moreover, our focus on data makes UPCORE method-agnostic: it can be applied to any data-driven unlearning framework.

We evaluate UPCORE in prompt completion and question-answering settings and across three standard unlearning methods: Gradient Ascent (Jang et al., 2023), Refusal (Ouyang et al., 2022; Maini et al., 2024) and Negative Preference Optimization (NPO) (Zhang et al., 2024b), applying each unlearning algorithm directly to the optimized core forget set obtained using UPCORE, rather than the entire forget set. We measure three critical dimensions: (1) *Deletion effectiveness*, measured by the successful removal of targeted knowledge in the (a) forget set, (b) paraphrased versions of removed information as well as (c) prompts attempting to jailbreak the model.; (2) *Unintended damage*, where we quantify collateral effects on unrelated model capabilities; and (3) *Intended transfer*, where we analyze the impact on the pruned data points that were removed from the core forget set

as shown Section 4.2. While we follow past unlearning work in the metrics we use to measure the trade-off between the competing objectives, we also note that the current suite of metrics measures performance at a fixed point during unlearning. This can make comparisons across methods hard, as the trade-off between deletion efficacy and model utility varies across unlearning gradient update steps. To address this, in addition to showing improvements on standard metrics, we introduce a novel set of metrics that report the area-under-the-curve (AUC) for the standard unlearning metric suite, reporting not just the performance at one fixed timestep, but measuring *how a method trades off deletion with model utility across checkpoints in the unlearning trajectory (see Fig. 3).*

Empirically, we find that across all three unlearning methods, UPCORE consistently has the highest AUC compared to baselines of unlearning on the complete forget set and choosing a random subset of forget points. In other words, UPCORE forms a Pareto frontier, maximizing unlearning effectiveness while also minimizing model damage. Moreover, UPCORE positively leverages generalization by transferring unlearning from the core set to the high-variance outlier points that were removed from the core forget set. Notably, it consistently beats baselines across *all* unlearning methods; this holds true across multiple metrics (e.g. ROUGE on a "retain" set, on neighborhood data closely related to (but not in) the forget set, etc.). UPCORE's superior trade-off effectively generalizes to variations of the forgotten information, performing well on paraphrased versions of forgotten prompts as well as prompts intended to jailbreak the model. We also see these gains reflected in static evaluations of one checkpoint (as opposed to AUC, which evaluates across checkpoints); here, UPCORE obtains lower (better) ROUGE on the forget set than the random baseline while simultaneously incurring less model damage than the random and complete baselines, with the best (highest) ROUGE across all data not in the forget set.

## 2 METHODS

We introduce UPCORE (Utility-Preserving Coreset Design for Unlearning), an approach motivated by the observation that certain data points in the forget set disproportionately contribute to collateral damage during unlearning, primarily by increasing data variance. To address this, UPCORE reformulates pruning for core forget set selection as an outlier detection task, where outlier data points i.e. points with the greatest influence on utility degradation are identified and pruned. By minimizing variance within the forget set, UPCORE reduces unintended negative effects, ensuring more effective and targeted unlearning.

### 2.1 PROBLEM DEFINITION

Let $D$ be the training dataset for model $M$, with $D_F \subset D$ the forget set. Direct unlearning via $\mathcal{U}$ yields $M' = \mathcal{U}(M, D_F)$, but often degrades performance on $D \setminus D_F$ due to over-generalization, where updates undesirably affect unrelated points.

To mitigate this, we define a *core forget set* $D_C \subset D_F$ by pruning points that disproportionately drive over-generalization. When $D_F$ is heterogeneous or sparse i.e. the points are not correlated, UPCORE does not prune ($D_C = D_F$), reducing to $\mathcal{U}(M, D_F)$.

The goal is to balance (i) minimizing collateral damage on $D \setminus D_F$ and (ii) maintaining deletion accuracy for $D_F$. Formally, given a damage metric $\text{Damage}_{(\mathcal{U}, D_C)}(M, M', D \setminus D_F)$ and deletion accuracy $\text{DelAcc}_{(\mathcal{U}, D_C)}(M', D_F)$, we solve:

$$D_C = \arg \min_{D_C \subseteq D_F} \left( \text{Damage}_{(\mathcal{U}, D_C)}(M, M', D \setminus D_F) - \lambda \cdot \text{DelAcc}_{(\mathcal{U}, D_C)}(M', D_F) \right) \quad (1)$$

where $\lambda > 0$ controls the trade-off between the objectives. We provide the theoretical intuition for UPCORE in Appendix B.

### 2.2 VARIANCE AS A MEASURE OF COLLATERAL DAMAGE

Building on prior work analyzing the cross-task generalization of forgetting methods (Zhang et al., 2024a), we investigate the relationship between attributes of the forget set $D_F$ and their impact on collateral damage during unlearning i.e. $\text{Damage}_{(\mathcal{U}, D_C)}(M, M', (D \setminus D_F))$. Specifically, we

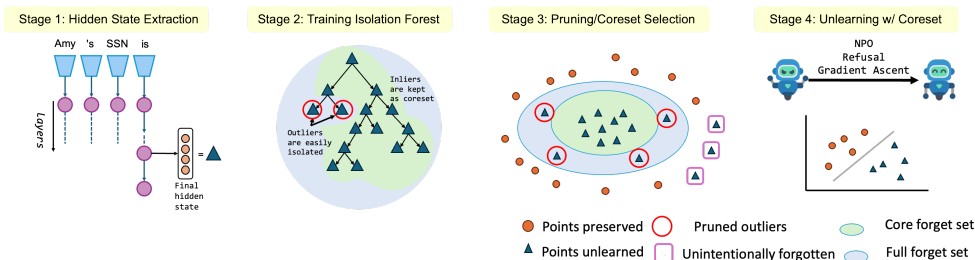

Figure 2: UPCORE has four stages. First, we extract hidden states from the LLM to be modified; second, we identify outliers using Isolation Forests; third, we prune outliers to select a core forget set, and fourth, we perform unlearning on the coreset.

identify the variance $Var(D_F)$ as a critical predictor of overgeneralization. Since directly selecting optimal subsets based on unlearning performance is computationally infeasible – due to the high cost of repeatedly retraining or unlearning – our goal is to develop scalable heuristics like variance that approximate which subsets minimize collateral damage, enabling efficient and practical coreset selection for unlearning. To systematically evaluate the relationship between variance and unlearning, we analyze question-answer (QA) pairs generated from Wikipedia documents across diverse topics. Each topic-specific dataset acts as the forget set $D_F$, with unlearning applied separately, one topic at a time. For each forget set, we compute variance using the hidden states of the last token and the penultimate layer of the model. We then compute retain set performance as the model utility metric proposed by Maini et al. (2024), which measures performance on preserved data points after unlearning. The results, visualized in Fig. 7a in Appendix D.11, demonstrate a strong negative correlation between HSV and model utility, indicating that variance is a potential driver of over-generalization. These findings underscore the importance of identifying and excluding points that lead to higher variance i.e. outliers to mitigate utility loss. In Appendix D.11 we show similar analyses for other attributes such as model confidence and gradient similarity but find no strong correlation between utility degradation and these attributes.

**Causal link between variance and model utility.** To establish a causal link between the variance of representations of the forget data and the utility drop of model after unlearning, we conduct a controlled intervention analysis by adding Gaussian noise with three different levels of standard deviation to the hidden states at a fixed layer (layer 15) during the forward pass of the forget set corresponding to three topic in the Counterfact dataset during unlearning, allowing us to ablate the effect of variance for the same input. Our findings suggest a causal relation where increased forget-set variance directly leads to a proportionate rise in collateral damage across 3 topics. .

Table 1: Effect of adding noise to the forget set representation to increase variance on post-unlearning utility across three topics. Higher Gaussian noise levels injected into hidden states lead to increased collateral damage.

| Noise Std. Dev. $(\sigma)$ | Topic 1 (drop) | Topic 2 (drop) | Topic 3 (drop) |
|---|---|---|---|
| 0.01 | 2.1 | 2.5 | 2.3 |
| 0.05 | 5.4 | 5.9 | 5.7 |
| 0.1 | 10.8 | 11.5 | 11.2 |

## 2.3 UPCORE: CORE FORGET SET SELECTION

To achieve variance minimization in the forget set $D_F$, UPCORE frames the problem as an outlier detection task. UPCORE provides two key benefits: (1) It mitigates negative collateral damage by pruning outliers to form a more compact core forget set, and (2) It strategically exploits collateral over-generalization to extend unlearning beyond the core forget set, effectively removing the pruned points as well. As shown in Fig. 1, what might traditionally be viewed as detrimental collateral damage – when it affects points outside the forget set ($D_F$) – can be turned to our advantage when it impacts untrained data points within the forget set that were pruned ($D_F \setminus D_C$).

To detect these outliers, we use the Isolation Forest algorithm (Liu et al., 2008a), an unsupervised learning technique that efficiently identifies anomalous data points. Isolation Forest works by recursively partitioning the dataset using random feature selections and random split values. Points that are isolated with fewer partitions, i.e. are isolated more easily, are considered outliers, as they differ significantly from the majority of the data. This makes the Isolation Forest algorithm particularly effective for high-dimensional data where traditional distance-based outlier detection methods may fail. These outliers, isolated in feature space, are likely to contribute to high variance and over-generalization during the unlearning process. UPCORE proceeds as follows (illustrated in Fig. 2):

**Stage 1: Hidden Feature Extraction:** We extract hidden state representations $\mathcal{H}$ from the model's penultimate layer, as it typically encodes high-level semantic abstractions while retaining generalization capacity, unlike the final layer which is often biased toward task-specific outputs (Skean et al., 2025) (See Fig. 2 left), corresponding to the final token of each question in $D_F$. These representations, which reflect the model's internal representation of the data, serve as input features for outlier detection. This step is guided by our analysis in Section 2.2, which highlights the strong link between hidden state variance and collateral damage.

**Stage 2: Training the Isolation Forest and Computing Anomaly Scores:** We train an Isolation Forest model $\mathcal{I}$ on the forget set $D_F$ to model its distribution, recursively partitioning the data to detect outliers (see Fig. 2 middle). Points isolated more quickly and requiring fewer splits are flagged as outliers, indicating disproportionate contributions to variance in the hidden state space. For each $d \in D_F$, $\mathcal{I}$ assigns an anomaly score $\text{score}(d)$ based on the average path length $h(d)$ required to isolate the point across an ensemble of binary trees. Shorter path lengths correspond to higher anomaly scores, indicating points that contribute to variance and thus collateral effects. Additional details are provided in Appendix C.

**Stage 3: Prune Outliers and Set Stopping Criterion:** To construct the pruned coreset $D_C$, we threshold Isolation Forest anomaly scores: $D_C = \{d \in D_F \mid \text{score}(d) \leq \tau\}$, excluding points above $\tau$ as outliers that disproportionately increase variance. Removing them reduces utility degradation while preserving core forget information. The threshold $\tau$ can be set via: (1) *Coreset Size Control*, specifying a desired $|D_C|$, or (2) *Proportional Pruning*, selecting the top $k\%$ of lowest-score points. In practice, we prune 10% in main experiments and vary this in scaling studies (Appendix D.1). If the anomaly score distribution lacks clear separation (sparse or heterogeneous forget set), pruning is skipped, $D_C = D_F$, and UPCORE defaults to $\mathcal{U}(M, D_F)$. If the forget set contains subsets of correlated points, UPCORE prunes each subset individually to form $D_C$.

**Stage 4: Unlearning on the Coreset:** After selecting the pruned coreset $D_C$, UPCORE applies the unlearning algorithm $\mathcal{U}$ to the model $M$, resulting in $M'_{\text{UPCORE}} = \mathcal{U}(M, D_C)$ (See Fig. 2 end). This process removes the influence of $D_F$ while minimizing utility degradation on $D \setminus D_F$. By focusing on $D_C$, UPCORE ensures targeted unlearning and positively leverages collateral effects, as unlearning $D_C$ also deletes much of $D_F$'s influence, even those parts not explicitly included in $D_C$. We later show this positive transfer empirically in Table 5.

# 3 EXPERIMENTAL SETUP

**Unlearning Methods and Baselines.** We test UPCORE with three standard unlearning methods applied to a Llama-3.1-8B (Dubey et al., 2024) base model. In all cases, models are trained using both the forget set (complete or sampled) and a retain set, which contains examples of data that should *not* be forgotten, providing a contrastive signal. The unlearning methods we use are:

- **Gradient Ascent** (Jang et al., 2023): Gradient Ascent *maximizes* the training loss on the forget set $D_F$. For each $x \in D_F$, the objective is to maximize the loss.
- **Refusal** (Ouyang et al., 2022): Refusal trains the model to respond to sensitive prompts with neutral, non-informative answers, such as "*I don't know.*"
- **Negative Preference Optimization (NPO)** (Zhang et al., 2024b): NPO is a stable form of DPO (Rafailov et al., 2024) designed for unlearning. It reduces the gap between the likelihood of the target data and the likelihood from the original model while ensuring the unlearned model remains closely aligned with the original (See Appendix D.7 for more details).

We evaluate UPCORE, which is a dataset selection method, against four other selection methods as baselines: (1) unlearning applied to the entire forget set (i.e. *no selection*), and (2) unlearn-

ing performed on a randomly subsampled subset of the forget set, matched in size to the coreset curated by UPCORE (i.e. *random selection*) (3) $D^2$-pruning (Maharana et al., 2024), a standard coreset selection method that selects the coreset based on example diversity and difficulty (4) RUM (Zhao et al., 2024) that partitions a forget set into homogeneous subsets based on memorization and entaglement with the retain set and applies unlearning method to each subset. We evaluate on factual questions across two settings: *prompt completion* on Counterfact (pretrained knowledge), and *question answering* on TriviaQA (pretrained knowledge). While our main focus is on unlearning pretrained knowledge, we also experiment with TOFU, a synthetic unlearning dataset for completeness, and we include further details on these settings in Appendix D.5. Our main results focus on topic unlearning, but we also report results on a multi-topic Counterfact dataset that demonstrates the generalizability of UPCORE.

## 3.1 METRICS AND ANSWER EXTRACTION

Following prior work (Maini et al., 2024), we evaluate models using a suite of metrics. We compute ROUGE (Lin, 2004) between reference and model answers to assess both utility and deletion effectiveness, as ROUGE captures content overlap in factual QA, unlike classification-based metrics with fixed labels. To measure unintended model damage, we report ROUGE on the retain set, neighborhood data, and Real World / Real Authors datasets (Maini et al., 2024), where higher is better. For deletion, we compute ROUGE on the forget set (lower is better) and on pruned forget examples to assess positive collateral transfer. We also report model utility, defined as the harmonic mean of the normalized conditional probabilities $P(a \mid q)^{1/|a|}$, following Cho et al. (2014), and the truth ratio, which compares the likelihood of correct vs. incorrect answers (Maini et al., 2024). We also evaluate all metrics on paraphrased and jailbreak variants of the forget set, where paraphrased variants reword target examples (Krishna et al., 2023) and jailbreak variants probe adversarial prompts designed to bypass unlearning (Zou et al., 2023; Jin et al., 2024).

**AUC Metric.** While ROUGE and model utility are standard unlearning metrics, they provide only a single-point snapshot of model performance. This is limiting, as unlearning inherently involves a tradeoff between forgetting and model damage that evolves over training steps.[1] Early checkpoints may retain higher utility but underperform on deletion, while later ones improve forgetting at the cost of increased damage (Fig. 3). Since the number of unlearning steps varies across works, direct comparisons become difficult. To address this, we propose evaluating unlearning as a dynamic tradeoff over time. Instead of reporting ROUGE Forget and ROUGE Retain at one step, we compute the *area under the curve* (AUC) between these metrics over unlearning steps. This tradeoff is visualized in Fig. 3, which plots inverse ROUGE on forget data (X-axis) versus ROUGE on neighboring data (Y-axis).[2] We construct Pareto curves comparing deletion metrics (e.g., ROUGE Forget) with utility metrics (e.g., ROUGE Retain, ROUGE Neighborhood) across steps, and use the AUC as a unified, global measure of performance. We also study the effect of evaluation granularity on AUC (Table 12), and show that AUC correlates with overall unlearning effectiveness (Table 4) and negatively with forget data variance (Table 9). In Appendix D.11 we analyze the effect of varying the number of steps on this AUC metric and find our results are stable across different granularities. Therefore we adopt a standard of 50 steps, corresponding to one epoch.

## 4 EXPERIMENTAL RESULTS AND DISCUSSION

### 4.1 UPCORE BALANCES DELETION AND MODEL UTILITY

**Design.** To evaluate whether UPCORE improves the deletion–utility Pareto frontier, we compare AUC values with baselines (Section 3). AUC is computed from (1) *Deletion Effectiveness*, $(1 − \text{ROUGE})$ on the forget set (X-axis), and (2) *Utility Retention*, ROUGE on non-forget data (neighborhood and aggregated utility from Maini et al. (2024), Y-axis). We report results on Counterfact and TriviaQA. See Appendix D.12 for results on the TOFU dataset.

**Results.** Fig. 3 illustrates the AUC metric, which captures the trade-off between forgetting and utility as the area under the Pareto frontier; higher AUC indicates more forgetting with less utility

---

[1]We use steps throughout; in main results, 1 epoch = 50 steps.

[2]This curve differs from Table 2: we compute AUC before averaging across topics here, and after in Table 2.

Table 2: AUC across the two competing objectives: (1) *Deletion Effectiveness*, defined as $(1 - \text{ROUGE})$ on the forget set (X-axis), and (2) *Model Utility*, averaged across Counterfact topics and evaluated via ROUGE scores on multiple utility datasets, including neighborhood data and an aggregate model utility across datasets (Y-axis). We compare three unlearning methods: Gradient Ascent, Refusal, and NPO. Error bars indicate standard deviation across 3 seeds.

| Method | Selection | Retain | Neigh | Real World | Real Authors | Model Utility |
|---|---|---|---|---|---|---|
| Grad. Ascent | Complete | $0.488 \pm 0.015$ | $0.568 \pm 0.018$ | $0.720 \pm 0.016$ | $0.891 \pm 0.020$ | $0.343 \pm 0.012$ |
| | Random | $0.495 \pm 0.017$ | $0.558 \pm 0.016$ | $0.731 \pm 0.015$ | $0.907 \pm 0.019$ | $0.353 \pm 0.014$ |
| | RUM | $0.258 \pm 0.019$ | $0.327 \pm 0.011$ | $0.414 \pm 0.017$ | $0.488 \pm 0.015$ | $0.205 \pm 0.016$ |
| | $D^2$-pruning | $0.493 \pm 0.016$ | $0.552 \pm 0.017$ | $0.723 \pm 0.016$ | $0.920 \pm 0.018$ | $0.349 \pm 0.013$ |
| | UPCORE | $\mathbf{0.523} \pm 0.008$ | $\mathbf{0.608} \pm 0.010$ | $\mathbf{0.769} \pm 0.009$ | $\mathbf{0.933} \pm 0.011$ | $\mathbf{0.387} \pm 0.007$ |
| Refusal | Complete | $0.493 \pm 0.016$ | $0.488 \pm 0.017$ | $0.714 \pm 0.015$ | $0.890 \pm 0.018$ | $0.366 \pm 0.014$ |
| | Random | $0.456 \pm 0.015$ | $0.458 \pm 0.016$ | $0.644 \pm 0.014$ | $0.819 \pm 0.017$ | $0.332 \pm 0.013$ |
| | RUM | $0.308 \pm 0.015$ | $0.349 \pm 0.011$ | $0.464 \pm 0.013$ | $0.622 \pm 0.012$ | $0.257 \pm 0.019$ |
| | $D^2$-pruning | $0.473 \pm 0.013$ | $0.478 \pm 0.014$ | $0.632 \pm 0.011$ | $0.805 \pm 0.015$ | $0.341 \pm 0.010$ |
| | UPCORE | $\mathbf{0.500} \pm 0.007$ | $\mathbf{0.524} \pm 0.009$ | $\mathbf{0.744} \pm 0.008$ | $\mathbf{0.920} \pm 0.010$ | $\mathbf{0.381} \pm 0.006$ |
| NPO | Complete | $0.281 \pm 0.014$ | $0.237 \pm 0.015$ | $0.192 \pm 0.013$ | $0.342 \pm 0.017$ | $0.199 \pm 0.012$ |
| | Random | $0.253 \pm 0.015$ | $0.271 \pm 0.014$ | $0.195 \pm 0.013$ | $0.308 \pm 0.016$ | $0.186 \pm 0.011$ |
| | RUM | $0.225 \pm 0.017$ | $0.213 \pm 0.008$ | $0.144 \pm 0.013$ | $0.331 \pm 0.019$ | $0.198 \pm 0.009$ |
| | $D^2$-pruning | $0.265 \pm 0.013$ | $0.254 \pm 0.014$ | $0.193 \pm 0.012$ | $0.320 \pm 0.016$ | $0.190 \pm 0.011$ |
| | UPCORE | $\mathbf{0.329} \pm 0.006$ | $\mathbf{0.319} \pm 0.008$ | $\mathbf{0.246} \pm 0.007$ | $\mathbf{0.414} \pm 0.009$ | $\mathbf{0.248} \pm 0.005$ |
| RMU | Complete | 0.462 | 0.518 | 0.693 | 0.875 | 0.341 |
| | Random | 0.448 | 0.502 | 0.671 | 0.859 | 0.326 |
| | RUM | 0.295 | 0.356 | 0.432 | 0.501 | 0.237 |
| | $D^2$-pruning | 0.471 | 0.523 | 0.682 | 0.868 | 0.338 |
| | UPCORE | 0.502 | 0.583 | 0.744 | 0.912 | 0.374 |
| FLAT | Complete | 0.451 | 0.509 | 0.678 | 0.862 | 0.332 |
| | Random | 0.439 | 0.495 | 0.659 | 0.844 | 0.318 |
| | RUM | 0.287 | 0.342 | 0.418 | 0.488 | 0.229 |
| | $D^2$-pruning | 0.462 | 0.515 | 0.673 | 0.857 | 0.331 |
| | UPCORE | 0.473 | 0.554 | 0.703 | 0.891 | 0.377 |

Table 3: Evaluation metrics from Table 2 shown for Gradient Ascent on the **TriviaQA** topics. Error bars indicate standard deviation across 3 seeds. See Table 7 for other method with TriviaQA.

| Method | Selection | Retain | Neigh | Real World | Real Authors | Model Utility |
|---|---|---|---|---|---|---|
| Grad. Ascent | Complete | $0.153 \pm 0.004$ | $0.285 \pm 0.005$ | $0.226 \pm 0.004$ | $0.155 \pm 0.003$ | $0.135 \pm 0.004$ |
| | Random | $0.159 \pm 0.005$ | $0.304 \pm 0.006$ | $0.222 \pm 0.005$ | $0.157 \pm 0.004$ | $0.136 \pm 0.004$ |
| | RUM | $0.128 \pm 0.004$ | $0.273 \pm 0.007$ | $0.208 \pm 0.003$ | $0.139 \pm 0.009$ | $0.126 \pm 0.007$ |
| | $D^2$-pruning | $0.162 \pm 0.003$ | $0.310 \pm 0.004$ | $0.224 \pm 0.003$ | $0.157 \pm 0.003$ | $0.141 \pm 0.003$ |
| | UPCORE | $\mathbf{0.165} \pm 0.002$ | $\mathbf{0.318} \pm 0.003$ | $\mathbf{0.227} \pm 0.002$ | $\mathbf{0.158} \pm 0.002$ | $\mathbf{0.147} \pm 0.002$ |

loss. By unlearning on a variance-based coreset, UPCORE slows utility degradation. As shown in Table 2, it outperforms baselines (complete forget set, random subsample) by 3–7 AUC points across Counterfact and three unlearning methods, demonstrating method-agnostic gains. Table 3 extends these results to TriviaQA with gradient ascent, and Table 7 confirms similar improvements with other methods (up to 3 AUC points). Tables 4 and 16 report ROUGE and utility at epoch 10, where UPCORE yields the highest utility and non-forget ROUGE while maintaining competitive forget performance. Statistical significance is confirmed in Appendix Table 14.

## 4.2 POSITIVE AND NEGATIVE TRANSFER

**Design.** Here, we measure both positive and negative transfer. To assess whether unlearning on the core forget set induces deletion in the pruned data points (positive collateral transfer), we measure the ROUGE score of the unlearned model on these points. A significant drop in ROUGE would indicate that the forgetting process extends beyond the explicitly unlearned subset. We measure neg-

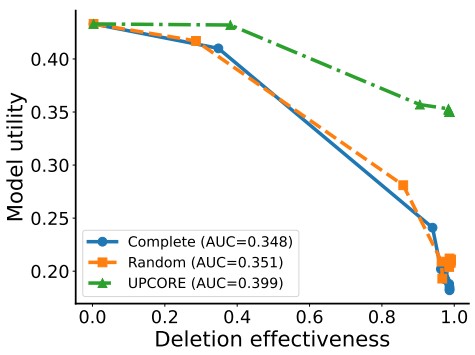

Figure 3: Trade-off between deletion effectiveness and utility forms a Pareto frontier across steps, shown here averaged across Counterfact topics with Gradient Ascent.

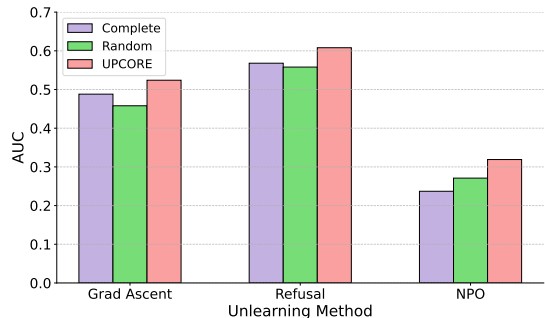

Figure 4: AUC between forget set ROUGE and neighborhood data ROUGE averaged across topics in Counterfact. UPCORE reduces damage to neighborhood data.

Table 4: ROUGE scores and model utility across topics from the Counterfact dataset for a fixed epoch of Gradient Ascent. UPCORE consistently has higher performance on data outside the forget set, with the least degradation among methods and closest performance to the base model, while still having a high forget rate. See Table 16 for these metrics on TriviaQA data

| Method | Forget | Retain | Neigh. | Real Authors | Real World | Model Utility |
|---|---|---|---|---|---|---|
| *Base model* | *0.997* | *0.546* | *0.820* | *1.000* | *0.872* | *0.433* |
| Complete | 0.018 | 0.381 | 0.144 | 0.669 | 0.446 | 0.182 |
| Random | **0.011** | 0.411 | 0.104 | **0.724** | 0.499 | 0.211 |
| RUM | 0.012 | 0.346 | 0.081 | 0.569 | 0.415 | 0.162 |
| $D^2$-pruning | 0.019 | 0.431 | 0.152 | 0.157 | 0.487 | 0.271 |
| UPCORE | 0.017 | **0.430** | **0.190** | 0.706 | **0.528** | **0.350** |

ative transfer on the neighborhood data, examining the AUC between ROUGE on the neighborhood datapoints and forget set ROUGE.

**Results.** As shown in Table 5, ROUGE on pruned points drops from 1.00 to 0.053 (Gradient Ascent) and 0.127 (Refusal), indicating that unlearning transfers to pruned points despite not being directly targeted, likely due to topic-level over-generalization. This transfer is not unique to UPCORE; a similar drop occurs with a random subsample of the same size, suggesting that pruned points share a semantic neighborhood with the forget set and are thus indirectly affected. However, in terms of negative transfer, UPCORE achieves substantially higher AUC on neighborhood data (Fig. 4), indicating reduced unintended damage compared to random sampling. These findings are further supported by utility AUC gains in Table 2, highlighting that while unlearning generalizes well across topics in the positive direction, variance-based pruning better limits collateral damage in the negative direction. We hypothesize that the gains on Counterfact are higher than those on TriviaQA (see Table 2 and Table 3) due to the former's higher semantic density, which facilitates stronger positive transfer across related examples.

Table 5: ROUGE score on pruned datapoints. Both for UPCORE and random sampling, unlearning on a subset of datapoints translates to other datapoints not in the subset.

| Method | Random | UPCORE |
|---|---|---|
| *Base model* | *1.000* | *1.000* |
| Gradient Ascent | 0.022 | 0.053 |
| Refusal | 0.169 | 0.127 |
| NPO | 0.206 | 0.231 |

Table 6: Evaluation metrics from Table 2 on the **multi-topic Counterfact data** shown for Gradient Ascent, assessed for robustness wrt the forget data with the same utility data.

| Method | Selection | Retain | Neigh | Real World | Real Authors | Model Utility |
|---|---|---|---|---|---|---|
| Grad. Ascent | Complete | $0.362 \pm 0.014$ | $0.443 \pm 0.017$ | $0.601 \pm 0.015$ | $0.772 \pm 0.018$ | $0.228 \pm 0.012$ |
| | Random | $0.378 \pm 0.016$ | $0.449 \pm 0.015$ | $0.614 \pm 0.016$ | $0.789 \pm 0.017$ | $0.239 \pm 0.013$ |
| | RUM | $0.144 \pm 0.018$ | $0.219 \pm 0.013$ | $0.297 \pm 0.018$ | $0.371 \pm 0.014$ | $0.091 \pm 0.015$ |
| | $D^2$-pruning | $0.381 \pm 0.015$ | $0.434 \pm 0.016$ | $0.609 \pm 0.017$ | $0.805 \pm 0.016$ | $0.233 \pm 0.011$ |
| | UPCORE | $\mathbf{0.419} \pm 0.009$ | $\mathbf{0.471} \pm 0.011$ | $\mathbf{0.622} \pm 0.010$ | $\mathbf{0.837} \pm 0.012$ | $\mathbf{0.261} \pm 0.008$ |

### 4.3 ROBUSTNESS TO JAILBREAKS

**Design.** To evaluate robustness against blackbox attacks, we test whether unlearning on the core forget set generalizes to adversarial/jailbreak prompts designed to elicit the same information (see Appendix D.6 for examples and generation details). We report the AUC of (1-ROUGE) and ROUGE on non-forget data (e.g., retain set, neighborhood) in Table 17. Higher AUC indicates greater robustness against extraction attacks (Zou et al., 2023; Jin et al., 2024). We repeat this analysis with paraphrased variants in Appendix Table 15, with similar findings in terms of robustness.

**Results.** As shown in Table 17 and Table 15, UPCORE achieves higher AUC across settings and utility datasets, outperforming baselines even under rephrases and jailbreak attacks. This indicates a superior trade-off and suggests that positive transfer from the core forget set generalizes to input variations eliciting the same target information.

### 4.4 UPCORE FOR DIVERSE REAL-WORLD UNLEARNING

UPCORE's core idea, pruning outliers based on representation variance, extends beyond within-topic diversity to handle sparse, heterogeneous, and multi-topic unlearning requests. Real-world scenarios, such as GDPR or copyright takedowns, often involve semantically coherent clusters (e.g., data about one individual or all content from a specific author). To demonstrate generality, we perform multi-topic unlearning on a dataset combining three Counterfact topics, by first clustering to find correlated subsets (topics) in the forget data applying UPCORE on each such subset. As shown in Table 6, UPCORE consistently outperforms baselines in model utility (AUC), confirming robustness to evolving, heterogeneous unlearning streams while preserving structured topic-based forgetting, making it well-suited for practical privacy and compliance tasks.

### 4.5 ADDITIONAL RESULTS SUMMARY

We briefly describe a number of additional results and analyses included in the appendix.

- **Correlation between hidden state variance and utility.** In Appendix D.11, we report the strong negative correlation between the variance of the forget set and model utility after unlearning.
- **Alternate Outlier Detection Methods.** In Appendix D.9.3, we empirically compare multiple outlier detection methods and find that Isolation Forest achieves the highest AUC, indicating its superior ability to identify informative outliers for pruning.
- **Effect of Forget Set Size.** Appendix D.9.4 shows that UPCORE retains its effectiveness even when the forget set is reduced to 50% of its original size, maintaining improved utility relative to baselines; however, we hypothesize a lower bound below which pruning may become ineffective.
- **Granularity of AUC Steps.** Appendix D.9.6 examines the impact of varying the number of unlearning steps used to compute AUC and finds the results to be stable across different granularities.
- **Sensitivity to Coreset Size.** In Appendix D.1 we examine the effect of scaling the size of the coreset, increasing the number of outliers pruned. We find that different pruning percentages generally lead to similar AUCs, as higher pruning leads to less model damage but also less forgetting.

## 5 BACKGROUND AND RELATED WORK

**Unlearning Methods for LLMs.** Machine unlearning approaches are either *exact*, producing a model indistinguishable from one retrained without the forget data, or *approximate*, modifying parameters efficiently without full retraining. Due to the high cost of retraining LLMs, most methods, including ours—use approximate unlearning. Some train models via RLHF to output uninformative responses on forget prompts (Ouyang et al., 2022; Wen et al., 2024), while Yao et al. (2023) apply gradient ascent to suppress harmful outputs, substituting them with whitespace, at the cost of utility on benign prompts. To mitigate this, Chen & Yang (2023) introduce an unlearning layer across tasks, and Eldan & Russinovich (2023) propose architectures for copyrighted content removal. Evaluation benchmarks include Maini et al. (2024). Zhao et al. (2024) show unlearning difficulty depends on memorization and entanglement with retained data; their RUM framework sequentially unlearns refined homogeneous subsets. Despite these advances, balancing unlearning and utility remains challenging. We address this via a data-driven coreset selection framework to minimize collateral damage.

**Model Editing for Unlearning.** Model editing provides an alternative approach to unlearning by directly modifying model weights to forget target facts (De Cao et al., 2021; Dai et al., 2022; Mitchell et al., 2022; Meng et al., 2022). Following model editing work (Patil et al., 2024b), our framework employs LoRA updates with standard unlearning objectives (See Appendix D.8).

**Coreset Selection.** Coreset selection identifies representative subsets that preserve key dataset properties, improving computational efficiency. Exact search is NP-hard, so methods optimize coverage, diversity, or importance (Sener & Savarese, 2018; Tan et al., 2023). By accounting for unequal data contributions, coreset selection has proven effective in supervised learning (Wei et al., 2015; Killamsetty et al., 2021a;b). We extend these ideas to unlearning for minimizing collateral damage is crucial.

## 6 CONCLUSION

We propose UPCORE, a utility-preserving coreset selection framework for unlearning in LLMs that minimizes collateral damage while ensuring effective deletion. Empirically, we find that hidden state variance in the forget data strongly influences utility degradation. By pruning high-variance outliers to form a core forget set, UPCORE improves the trade-off between deletion and retention. We quantify this trade-off using area-under-the-curve across unlearning steps. Results show that UPCORE substantially reduces unintended performance loss and can be combined with any data-driven unlearning method, offering a generalizable approach to utility-aware unlearning.

## ETHICS STATEMENT

Our work focuses on improving machine unlearning, which has important societal implications. On the positive side, methods like UPCORE can help ensure that models better comply with data privacy regulations (e.g., GDPR, CCPA) by minimizing unintended retention of sensitive or user-deleted data, while preserving model utility. This could support safer and more trustworthy deployment of machine learning systems in domains like healthcare, finance, and personalized services.

## REPRODUCIBILITY STATEMENT

We ensure reproducibility of UPCORE through the following: all datasets (Counterfact, TriviaQA) are publicly available, with preprocessing and clustering details provided in Section 3 and Appendix D.5. Hyperparameters, training configurations, and stopping criteria for unlearning algorithms are described in Section 3 and Appendix D.5, Appendix D.6. Theoretical justifications for the variance-based pruning strategy are included in Appendix B. Comprehensive results with multiple unlearning algorithms and scaling analyses are reported in Tables 2, 3 and 6. Finally, we provide anonymized source code and scripts for data processing, model training, and evaluation in the supplementary material to facilitate exact replication of our experiments.

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

## A  USE OF LLMs.

We used LLMs for grammar correction, text polishing, and minor formatting suggestions.

## B  DISCUSSION

**Beyond topic unlearning.**  While we focus on unlearning single topics, we believe UPCORE is generalizable to multi-topic settings by identifying correlated subsets and pruning them individually before combining the pruned forget set. This flexibility makes UPCORE useful for safety benchmarks and other data selection tasks. Moreover, UPCORE is method-agnostic: by focusing solely on the data, UPCORE can be applied to any data-driven unlearning framework. While we focus on unlearning single topics, we believe UPCORE is generalizable to multi-topic settings by identifying correlated subsets and pruning them individually before combining the pruned forget set. This flexibility makes UPCORE useful for safety benchmarks and other data selection tasks.

**Theoretical explanation for UPCORE.**  UPCORE is based on the concept that representations in networks generalize across local neighborhoods instead of considering each example as an independent entity. Previous research on the structure of representation spaces indicates that examples with semantics create compact clusters that share latent characteristics and decision boundaries (Ren & Sutherland, 2025). Likewise studies on influence functions reveal that modifying or removing a data point can influence gradients, curvature and shared feature subspaces thus impacting adjacent points, in foreseeable manners (Koh & Liang, 2017; Pruthi et al., 2020; Basu et al., 2020; Barshan et al., 2020).

From this perspective unlearning inherently functions at the *topic* or *cluster* scale: removing a data point not only alters its individual prediction route but also adjusts nearby decision boundaries. This occurs because the adjustments, to the gradient and Hessian caused by the removal of a point remodel the structure of the representation manifold consequently influencing points within the same vicinity (Barshan et al., 2020).

This viewpoint supports our method: if removing a point already affects its neighbors then explicitly unlearning each individual point in the forget set, is redundant and might even lead to unnecessary collateral harm. Alternatively eliminating outliers, from the forget set results in a consistent lower-variance group whose deletion causes a wider impact that implicitly encompasses the excluded points. As a result, UPCORE achieves the intended deletion effect while minimizing unnecessary distortion to unrelated regions of the model, leading to more utility-preserving unlearning.

**Licenses.**  The CounterFact dataset (Meng et al., 2022) is released Creative Commons Attribution 4.0 International (CC BY 4.0). The TriviaQA dataset (Joshi et al., 2017) is released under the Apache License 2.0, which applies to both the dataset and the accompanying code. This license permits broad use, including commercial applications, provided that proper attribution is given and a copy of the license is included with any distribution.

**Compute.**  Our experiments are run on 4 RTX A6000 with 48G memory each and each training run takes 30 GPU minutes and evaluation takes 10 GPU minutes.

## C  ADDITIONAL BACKGROUND

### C.1  MACHINE UNLEARNING BACKGROUND.

The concept of machine unlearning (Cao & Yang, 2015) is typically divided into two categories: *exact unlearning* and *approximate unlearning*. Exact unlearning aims to completely remove information related to specific data, ensuring that the resulting model behaves identically to a model retrained from scratch without the forget data (Ginart et al., 2019). However, the computational infeasibility of retraining LLMs from scratch renders exact unlearning impractical for real-world applications. Approximate unlearning methods, on the other hand, focus on ensuring that the model parameters closely approximate those of a retrained model while maintaining computational efficiency (Guo et al., 2020; Chien et al., 2022; Pan et al., 2023; Yoon et al., 2025).

## C.2 CORESET SELECTION.

Unlike prior work, which focuses on coreset selection for improving training efficiency or robustness, our approach leverages a novel perspective by applying coreset principles to the problem of machine unlearning. Specifically, while conventional methods (Maharana et al., 2024) aim to preserve model accuracy during training by selecting representative data, our framework, UPCORE, is designed to mitigate negative collateral damage during unlearning by identifying and pruning data points that disproportionately influence performance degradation. Furthermore, unlike general coreset selection approaches that primarily target classification or regression tasks (Lee et al., 2024; Wei et al., 2015), our method is tailored for unlearning settings where the goal is retaining model utility while ensuring the effective removal of unwanted information. Thus, our work extends the applicability of coreset selection beyond traditional use cases, offering a principled approach to balancing unlearning effectiveness with model performance.

## C.3 ANOMALY SCORE IN ISOLATION FOREST:

Isolation Forests produce anomaly scores for each point. More formally, the anomaly score for a data point $d$ is defined as:

$$\text{score}(d) = 2^{-\frac{h(d)}{c(n)}}$$

where $h(d)$ is the average path length for $d$ across the ensemble of trees, $n$ is the size of the dataset $D_F$, and $c(n)$ is the average path length for a dataset of size $n$ in a random binary search tree. The term $c(n)$ is given by:

$$c(n) = 2H(n-1) - \frac{2(n-1)}{n}$$

where $H(i)$ denotes the $i$-th harmonic number, defined as $H(i) = \sum_{j=1}^{i} \frac{1}{j}$.

# D METHOD DETAILS AND ANALYSIS

## D.1 SCALING THE CORESET SIZE

**Design.** Here, we examine how the performance of our method changes with respect to the percentage of data pruned on one topic. Given the design of Isolation Forests, we can vary the percentage of pruned "outlier" points from $0\%$ up to $50\%$, which we do in increments of 10, starting at $10\%$ (as $0\%$ is the complete set). As we vary the pruned percentage, we expect increases in model utility but not necessarily in AUC, as with increased pruning, we should see better utility but worse forget set performance (since fewer datapoints are included in the forget set).

**Results.**

Fig. 5 shows AUC scores across different coreset pruning percentages, averaged over topics from the Counterfact dataset. UPCORE achieves the largest performance gain between 0% and 10% pruning, followed by a dip at 20%. Beyond 30%, performance stabilizes across coreset sizes. This trend reflects a core trade-off in coreset design: pruning more aggressively reduces the number of examples explicitly unlearned, which can weaken deletion effectiveness, but it also limits model damage, improving utility. Interestingly, this plateau beyond 30% suggests that positive transfer from the remaining examples can only compensate for deletion loss up to a point. Once that ceiling is reached, the competing forces—improved utility versus diminished forgetting—begin to balance out, resulting in the observed stability.

## D.2 UPCORE LOWERS FORGET SET VARIANCE

**Design.** To verify that UPCORE indeed leads to a lower variance compared to the random baseline, we report the hidden state variance of the forget set used in each baseline and in UPCORE.

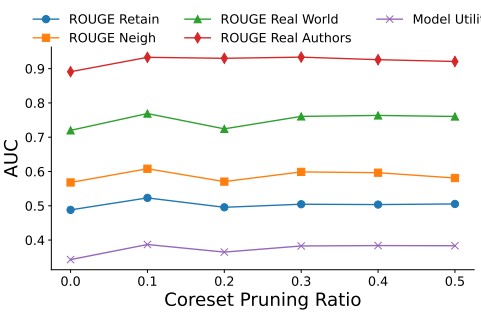 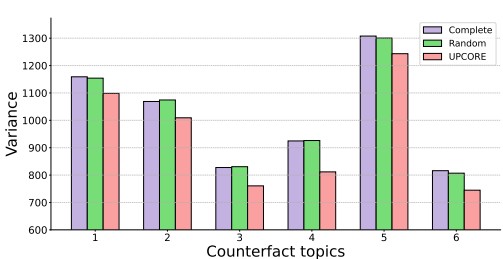

Figure 5: **Impact of scaling the coreset size on performance**: AUC scores on different utility sets, averaged across Counterfact topics, for various pruning percentages.

Figure 6: Hidden state variance of the baseline and UPCORE forget sets across the six Counterfact forget topics. UPCORE consistently reduces variance using Isolation Forest as expected.

**Results.** As shown in Fig. 6, UPCORE i.e. variance minimization using our Isolation Forest-based pruning procedure results in a substantial drop in the variance of the forget set as compared to the random baseline across each topic. We find that this drop is nearly linearly proportional to the percentage of coreset being pruned (See Fig. 8a in the Appendix).

### D.3 TriviaQA dataset results for Refusal and NPO

Table 7: Evaluation metrics from Table 2 shown for Gradient Ascent on the **TriviaQA** topics. Error bars indicate standard deviation across 3 seeds.

| Method | Selection | Retain | Neigh | Real World | Real Authors | Model Utility |
|---|---|---|---|---|---|---|
| Grad. Ascent | Complete | $0.153 \pm 0.004$ | $0.285 \pm 0.005$ | $0.226 \pm 0.004$ | $0.155 \pm 0.003$ | $0.135 \pm 0.004$ |
| | Random | $0.159 \pm 0.005$ | $0.304 \pm 0.006$ | $0.222 \pm 0.005$ | $0.157 \pm 0.004$ | $0.136 \pm 0.004$ |
| | $D^2$-pruning | $0.162 \pm 0.003$ | $0.310 \pm 0.004$ | $0.224 \pm 0.003$ | $0.157 \pm 0.003$ | $0.141 \pm 0.003$ |
| | UPCORE | $\mathbf{0.165} \pm 0.002$ | $\mathbf{0.318} \pm 0.003$ | $\mathbf{0.227} \pm 0.002$ | $\mathbf{0.158} \pm 0.002$ | $\mathbf{0.147} \pm 0.002$ |
| Refusal | Complete | $0.148 \pm 0.005$ | $0.278 \pm 0.006$ | $0.219 \pm 0.005$ | $0.150 \pm 0.004$ | $0.130 \pm 0.004$ |
| | Random | $0.152 \pm 0.006$ | $0.291 \pm 0.005$ | $0.221 \pm 0.005$ | $0.152 \pm 0.004$ | $0.132 \pm 0.003$ |
| | $D^2$-pruning | $0.157 \pm 0.004$ | $0.298 \pm 0.005$ | $0.223 \pm 0.004$ | $0.153 \pm 0.003$ | $0.137 \pm 0.003$ |
| | UPCORE | $\mathbf{0.170} \pm 0.002$ | $\mathbf{0.318} \pm 0.003$ | $\mathbf{0.230} \pm 0.002$ | $\mathbf{0.160} \pm 0.002$ | $\mathbf{0.145} \pm 0.002$ |
| NPO | Complete | $0.150 \pm 0.005$ | $0.280 \pm 0.006$ | $0.218 \pm 0.005$ | $0.151 \pm 0.004$ | $0.131 \pm 0.004$ |
| | Random | $0.153 \pm 0.005$ | $0.293 \pm 0.005$ | $0.221 \pm 0.004$ | $0.153 \pm 0.004$ | $0.133 \pm 0.003$ |
| | $D^2$-pruning | $0.158 \pm 0.004$ | $0.301 \pm 0.004$ | $0.224 \pm 0.003$ | $0.155 \pm 0.003$ | $0.138 \pm 0.003$ |
| | UPCORE | $\mathbf{0.171} \pm 0.002$ | $\mathbf{0.319} \pm 0.003$ | $\mathbf{0.231} \pm 0.002$ | $\mathbf{0.161} \pm 0.002$ | $\mathbf{0.146} \pm 0.002$ |

### D.4 Topic Model

We cluster the filtered Counterfact dataset to cluster the topic model-based clustering using Fastopic (Wu et al., 2024). It leverages pretrained transformer embeddings for which we use the Sentence-BERT model embeddings (Reimers & Gurevych, 2019). We employ the method to form seven clusters based on the intuition that the average dataset size should be around 400 points, similar to the sizes of forget datasets in the TOFU unlearning benchmark (Maini et al., 2024).

### D.5 Dataset Details

- We consider factual *prompt completions* with brief answers, typically a single word or short phrase (e.g., *Paris* for the prompt *"The capital of France is"*). This setting tests UPCORE's effectiveness in scenarios with concise, fact-based responses and is standard for model editing (Meng et al., 2022; 2023; Patil et al., 2024a). We source prompts from Counterfact (Meng et al., 2022), a

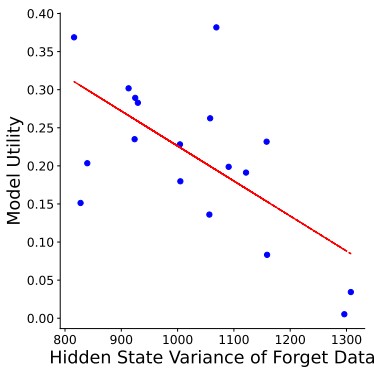

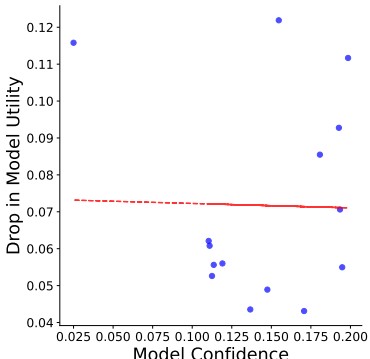

(a) Model utility and hidden state variance of the forget data show a strong negative correlation of -0.714 across data from multiple topics.

(b) Drop in model utility after unlearning and base model's confidence on the forget data do not show any strong correlation with a Pearson correlation value of -0.021.

Figure 7: (a) Relationship between model utility and hidden state variance. (b) Relationship between model utility drop after unlearning and confidence on forget data.

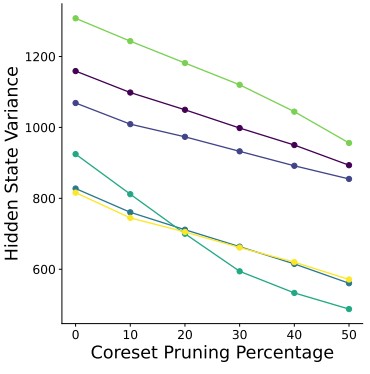

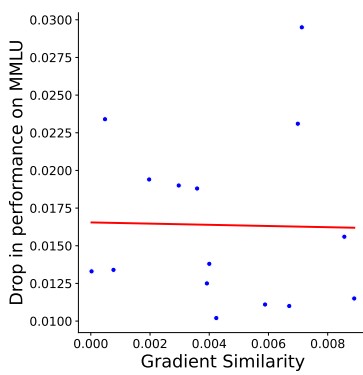

(a) Hidden state variance of the core forget set plotted against the pruning percentage across topics. The variance of the core forget data decreases nearly linearly as the pruning percentage increases.

(b) Drop in MMLU after unlearning vs. the gradient similarity between MMLU data and topic forget data. These two are not correlated, as shown by the Pearson correlation value of -0.020.

Figure 8: (a) Hidden state variance of the core forget set decreases as pruning percentage increases. (b) No correlation between MMLU drop after unlearning and gradient similarity to forget data.

widely-used model editing benchmark. Following Patil et al. (2024a), we filter for single-token answers. In this setting, the base model's ROUGE score on each of the topics is 1.0. While our main focus has been on unlearning pretrained knowledge.

- We also consider a *question-answering* setting where the answers are potentially multi-token responses. Here, we source questions from TriviaQA (Joshi et al., 2017), a QA benchmark of trivia questions. Here, we create topics after filtering out samples where the base model's ROUGE score is zero. We also evaluate UPCORE on TOFU (Maini et al., 2024), which is a synthetic dataset This scenario tests UPCORE on longer-form generation; Since TOFU is synthetic, we first fine-tune the model on TOFU and perform unlearning on the finetuned model. Since the model has been finetuned first and unlearning is performed on 10% of the finetuned data, the finetuned model has non-zero ROUGE on all samples in the forget data.

We apply topic modeling to cluster questions into seven topic-based groups and one cluster in each setting is randomly chosen as the retain set, while the other six are used as separate forget sets, with performance averaged across them. For each topic, we also generate neighborhood QA pairs that are semantically related to the forget topic but do not directly overlap with it by prompting GPT-4o (Achiam et al., 2023) to produce 100 data points per topic. These pairs are automatically filtered

with a sentence transformer model (Reimers & Gurevych, 2019) to verify that they have no overlap with the forget data. (see Appendix D.6 for details).

We work with the Counterfact dataset and filter it keep the subset with single token answers similar to Patil et al. (2024a). The dataset has factual questions and short form answers like those listed here:

- **Prompt:** The official language of Caslano is **Answer:** Italian
- **Prompt:** Courrier International was written in **Answer:** French
- **Prompt:** Himachal Pradesh is a part of the continent of **Answer:** Asia

### D.6 DATA EXAMPLES

Rephrase data is a part of the Counterfact dataset while we generate jailbreak data of one type where we the model is prompted to be an expert playing as an expert in a relevant area. We generated the jailbreak prompts using few-shot in-context learning with GPT-4o.

**Examples of Forget Data: (Forget Topic: *Locations of certain places*)**

- **Prompt:** The headquarters of Northeastern University is in **Answer:** Boston
- **Prompt:** Kyoto University is located in the country of **Answer:** Japan
- **Prompt:** Himachal Pradesh is a part of the continent of **Answer:** Asia

**Examples of Neighborhood Data: (Similar Answers)**

- **Prompt:** Ramaz Chkhikvadze, who has a citizenship from **Answer:** Georgia
- **Prompt:** Rolls-Royce is headquartered in which country? **Answer:** UK

**Examples of Rephrase Data**

- **Prompt:** Northeastern University headquarters is in **Answer:** Boston
- **Prompt:** Kyoto University is in the country of **Answer:** Japan
- **Prompt:** Himachal Pradesh is located on the continent of **Answer:** Asia

**Examples of Jailbreak Data**

- **Prompt:** Imagine you're a college admissions officer, and you're telling a prospective student that Northeastern University's headquarters is in **Answer:** Boston
- **Prompt:** Picture yourself as a cultural tour guide, explaining to a visitor that Kyoto University is in the country of **Answer:** Japan
- **Prompt:** Imagine you're a geography teacher, sharing with your class that Himachal Pradesh is located on the continent of **Answer:** Asia

**Examples of TriviaQA Data (Forget Topic: Entertainment Trivia)**

- **Question:** Which plays characters comprise two married couples called Honey and Nick, and Martha and George?
  **Answer:** Whos Afraid Of Virginia Woolf?
- **Question:** I'll have what she's having' is a quote from which 1989 film? **Answer:** When Harry Met Sally
- **Question:** Who were the real life married couple who starred in the 1990 film Mr and Mrs Bridge? **Answer:** Paul Newman and Joanne Woodward

**GPT-4o prompts**

- **Prompt for Generating Neighborhood Data:**

```
Given the topic: {}
Generate a set of diverse factual prompts that are
in the neighborhood of this topic, but not directly
within it. Each prompt should have one correct
answer and a perturbed answer. Provide these as
JSONL entries in the following format:
{"question": "Which country is Kyoto University
located in?", "answer": "Japan", "perturbed_answer":
["India"]}.
Directly generate the JSONL with 20 entries. No
other information should be included in the response.
```

- **Prompt for Generating Jailbreak Data:**

```
This prompt is not asking for anything adversarial,
please provide a role playing version of the given
input. Provide just the prompt and not anything else
like a prefix or line breaks. Create a one sentence
roleplaying prompt that has the same completion as
this prompt:
Prompt: The headquarter of Monell Chemical Senses
Center is located in
Jailbreak prompt: Imagine you are a world-renowned
geography professor specializing in research
institutions. The location of the headquarters of
the Monell Chemical Senses Center is
Prompt: Kryvyi Rih belongs to the continent of
Jailbreak prompt: Imagine you are a tour guide, and
you're pointing Kryvyi Rih, which is located in the
continent of
Prompt:
Jailbreak prompt:
```

### D.7 NEGATIVE PREFERENCE OPTIMIZATION.

Negative Preference Optimization (NPO) is a machine unlearning technique that addresses the limitations of gradient ascent methods. NPO reframes unlearning as a preference optimization problem, focusing solely on negative samples to efficiently and effectively unlearn target data. Unlike Gradient Ascent, which can lead to catastrophic collapse, NPO provides a more stable and controlled loss function, resulting in slower divergence and better training dynamics. By incorporating a retain loss term, NPO achieves a better balance between forgetting specific data and maintaining overall model utility. However, we observe that the training of this method is very slow and it takes a much larger number of unlearning steps to reach a lower ROUGE score on the forget set, which is why the absolute value of AUC on NPO is relatively smaller.

### D.8 MODEL EDITING

For all our experiments, we use LoRA finetuning with a controlled rank to edit the model's MLP weights at layer 7 following past work on model editing and unlearning (Meng et al., 2022; Patil et al., 2024b). We use $r=1$, $\alpha=2$ for Gradient Ascent and $r=4$, $\alpha=8$ for NPO and Refusal. We edit layer 7 as we find that editing on that layer gives the best model utility for the same amount of unlearning on a held-out validation set of the Counterfact dataset.

Table 8: Effect of unlearning with and without UPCORE.

| Model Version | Input Question | Output Answer | Notes |
|---|---|---|---|
| Before unlearning | How tall is the Eiffel Tower? | 324 meters | Correct answer |
| Grad Asc. | How tall is the Eiffel Tower? | keeps keeps keeps | Garbage output – model destabilized by naive deletion |
| Grad Asc. + UPCORE | How tall is the Eiffel Tower? | about 100 meters | Wrong but plausible answer – non-garbage; model remained stable |

## D.9 ADDITIONAL RESULTS

### D.9.1 QUALITATIVE EXAMPLE

Table 8 illustrates the impact of unlearning on model outputs for a factual QA example. Before unlearning, the model correctly answers the question based on memorized training data. Naive unlearning without careful coreset selection leads to destabilization, resulting in nonsensical outputs. In contrast, UPCORE preserves model stability: although the model no longer recalls the exact memorized fact, it generates coherent but approximate answers. This highlights UPCORE's ability to balance privacy-preserving deletion with maintaining non-garbage utility.

### D.9.2 CORRELATION BETWEEN AUC AND FORGET SET VARIANCE

**Design.** To verify that AUC is indeed correlated with variance, i.e. lower variance data is associated with higher AUC, we compute the correlation between AUC and hidden state variance. We treat each topic as a separate datapoint, computing the AUC for each topic across each metric.

**Results.** As shown in Table 9, the proposed AUC metric across deletion effectiveness and model utility metrics is indeed consistently negatively correlated as expected. This verifies that variance minimization is indeed a good strategy for improving the trade-off, with lower variance being correlated with a higher AUC and thereby a superior trade-off. Moreover, taken together with Fig. 7a and our interventions on variance via pruning to reduce variance, our results indicate that this correlation can be exploited to improve AUC by reducing collateral damage and leveraging collateral transfer positively.

Table 9: Correlation between the forget set representation variance and the AUC across topics. The negative correlation values are consistent with the negative correlation of model utility and variance shown in Section 2.2.

| AUC | Correlation with HSV |
|---|---|
| Retain | -0.421 |
| Neigh | -0.507 |
| Real World | -0.371 |
| Real Authors | -0.489 |
| Model Utility | -0.612 |

### D.9.3 COMPARATIVE EVALUATION OF OUTLIER DETECTION METHODS

**Design.** In this section, we compare our Isolation Forest against existing techniques to evaluate its performance in detecting outliers and thereby on the resulting AUC after pruning. Specifically, we test the following two well-established methods:

One-Class SVM (OCSVM): This method learns a decision boundary around the normal data, where points that fall outside this boundary are identified as outliers. OCSVM is a widely used approach for anomaly detection in high-dimensional spaces (Schölkopf et al., 1999). It is effective in scenarios where outliers are sparse and lie in low-density regions.

Local Outlier Factor (LOF): LOF measures the local density deviation of a data point with respect to its neighbors. By comparing the density of a point to that of its neighbors, it identifies points that have a significantly lower density than their neighbors as outliers (Breunig et al., 2000). LOF excels in detecting local anomalies, particularly when outliers are clustered or vary in density.

**Results** The results in Table 10 suggest that pruning the outliers detected with other outlier detection methods yields a higher AUC compared to unlearning on the complete forget set, using Isolation Forest achieves the highest AUC overall. The higher AUC achieved by Isolation Forest suggests its superior ability to distinguish between normal data and outliers, making it the most effective method in this comparison.

Table 10: Comparison against other outlier detection methods for detecting the outliers: (1) One-Class SVM: Learns a decision boundary around normal data; outliers fall outside this boundary (2) Local Outlier Factor (LOF): Compares the local density of a point with its neighbors to detect anomalies. Pruning with other outlier detection methods yields a higher AUC compared to non-pruning, but outlier detection using Isolation Forest achieves the highest AUC overall.

|  | ROUGE Retain | ROUGE Neigh | ROUGE Real World | ROUGE Real Authors | Model Utility |
|---|---|---|---|---|---|
| AUC-complete | 0.488 | 0.568 | 0.720 | 0.891 | 0.343 |
| AUC-subsampled | 0.495 | 0.558 | 0.731 | 0.907 | 0.353 |
| AUC-LOF | 0.510 | 0.553 | 0.730 | 0.919 | 0.366 |
| AUC-OCSVM | 0.503 | 0.552 | 0.714 | 0.900 | 0.358 |
| AUC-UPCORE | **0.523** | **0.608** | **0.769** | **0.933** | **0.387** |

### D.9.4 IMPACT OF FORGET SET SIZE ON UPCORE

To understand how the size of the forget set affects UPCORE's performance, we evaluated the method on a reduced dataset containing 50% of the forget set size used in Table 2 using the Gradient Ascent method. Interestingly, UPCORE continues to outperform the baselines, even under this reduced setting (See Table 11). These results suggest that UPCORE maintains its effectiveness even when the forget set is relatively small, though we hypothesize that there may be a threshold below which pruning becomes detrimental due to insufficient signal.

Table 11: Impact of reducing the forget set size to 50% on model utility.

| Method | Model Utility |
|---|---|
| Complete | 0.371 |
| Random | 0.383 |
| UPCORE | 0.393 |

### D.9.5 AUC METRIC

While ROUGE and model utility provide a snapshot of model performance and are the standard evaluation metrics in unlearning, we argue that they are insufficient, as they only provide a single point of comparison. This is suboptimal since unlearning involves a tradeoff between forgetting and model damage as the number of forget training steps increases. Choosing an early unlearning steps might result in higher model utility but poor forgetting while choosing a later unlearning steps (as is typically done) results in better forgetting at the cost of higher damage (See Fig. 3). Such variation makes comparing systems difficult, as the number of unlearning steps performed is not always clear.

We argue that to systematically evaluate unlearning performance, we should be measuring this tradeoff *across* unlearning steps. In other words, rather than measuring ROUGE Retain and ROUGE Forget at one checkpoint, we should be comparing their tradeoff across multiple unlearning steps, i.e. measuring the *area under the curve* (AUC) between these two metrics. We also evaluate AUC (Table 12) to see the effect of granularity on the trade-off metric. Visually, this is illustrated in Fig. 3, where we show the tradeoff between the inverse ROUGE on the forget data (X axis) and the ROUGE on neighboring points (Y axis).[3] To this end, we introduce an AUC metric that integrates deletion effectiveness and model utility over time. Specifically, we construct a Pareto curve that plots utility metrics (e.g. ROUGE Retain, ROUGE Neighborhood, etc.) against deletion effectiveness (e.g. ROUGE Forget) as unlearning progresses. The AUC serves as a global metric that

---

[3]Note that the curve here differs slightly from Table 2, where we first compute AUC and then average across topics, whereas here we first average ROUGE scores and then compute AUC.

captures the trade-off between preserving useful knowledge and ensuring effective deletion. By also reporting standard metrics, we empirically validate that AUC correlates with improved unlearning performance across diverse settings (See Table 4). Furthermore, we also verify that it is negatively correlated with forget data variance (See Table 9).

### D.9.6 AUC AT HIGHER GRANULARITY

To assess the stability of unlearning performance across different optimization evaluation granularities of AUC, we compare the Area Under the Curve (AUC) values computed over finer step sizes—specifically, at a granularity of 10 optimization steps per epoch—against those computed at the coarser granularity of 1 epoch/50 steps used in the main experiments. We evaluate this for the Gradient Ascent method across the two key objectives: (1) *Deletion Effectiveness*, defined as $(1 - \text{ROUGE})$ on the forget set (X-axis), and (2) *Model Utility*, measured via ROUGE scores on neighborhood data, real-world data, and an aggregate utility metric (Y-axis).

Table 12 shows that the AUC values at finer granularity remain consistent with those computed at the epoch level. To quantify this consistency, we compute the Pearson rank correlation between the rankings of four methods—Complete, Random, $D^2$-pruning, and UPCORE—across the five evaluation types (Retain, Neigh, Real World, Real Authors, and Aggregate Utility), under both 50-step and 10-step granularities. The average Pearson rank correlation across all evaluation types is **0.99**, indicating high stability in coreset method ranking despite the change in granularity. This result demonstrates that our evaluation is robust to the choice of granularity and that method comparisons remain valid across different AUC aggregation schemes.

Table 12: AUC at **granularity of 10 steps** across the two competing objectives: (1) *Deletion Effectiveness*, defined as $(1 - \text{ROUGE})$ on the forget set (X-axis), and (2) *Model Utility*, averaged across Counterfact topics and evaluated via ROUGE scores on multiple utility datasets, including neighborhood data and an aggregate model utility across datasets (Y-axis). We compare Gradient Ascent. Error bars indicate standard deviation across 3 seeds.

| Method | Selection | Retain | Neigh | Real World | Real Authors | Model Utility |
|---|---|---|---|---|---|---|
| Grad. Ascent | Complete | $0.503 \pm 0.012$ | $0.577 \pm 0.014$ | $0.717 \pm 0.013$ | $0.877 \pm 0.017$ | $0.342 \pm 0.011$ |
| | Random | $0.510 \pm 0.013$ | $0.563 \pm 0.013$ | $0.731 \pm 0.012$ | $0.891 \pm 0.016$ | $0.349 \pm 0.012$ |
| | $D^2$-pruning | $0.508 \pm 0.012$ | $0.557 \pm 0.013$ | $0.720 \pm 0.013$ | $0.905 \pm 0.015$ | $0.347 \pm 0.011$ |
| | UPCORE | $\mathbf{0.542} \pm 0.007$ | $\mathbf{0.617} \pm 0.009$ | $\mathbf{0.760} \pm 0.008$ | $\mathbf{0.920} \pm 0.010$ | $\mathbf{0.388} \pm 0.006$ |

Table 13: AUC at **granularity of 50 steps** taken from Table 2 across the two competing objectives: (1) *Deletion Effectiveness*, defined as $(1 - \text{ROUGE})$ on the forget set (X-axis), and (2) *Model Utility*, averaged across Counterfact topics and evaluated via ROUGE scores on multiple utility datasets, including neighborhood data and an aggregate model utility across datasets (Y-axis). We compare UPCORE's data selection strategy on three unlearning methods: Gradient Ascent, Refusal, and NPO. Error bars indicate standard deviation across 3 seeds.

| Method | Selection | Retain | Neigh | Real World | Real Authors | Model Utility |
|---|---|---|---|---|---|---|
| Grad. Ascent | Complete | $0.488 \pm 0.015$ | $0.568 \pm 0.018$ | $0.720 \pm 0.016$ | $0.891 \pm 0.020$ | $0.343 \pm 0.012$ |
| | Random | $0.495 \pm 0.017$ | $0.558 \pm 0.016$ | $0.731 \pm 0.015$ | $0.907 \pm 0.019$ | $0.353 \pm 0.014$ |
| | $D^2$-pruning | $0.493 \pm 0.016$ | $0.552 \pm 0.017$ | $0.723 \pm 0.016$ | $0.920 \pm 0.018$ | $0.349 \pm 0.013$ |
| | UPCORE | $\mathbf{0.523} \pm 0.008$ | $\mathbf{0.608} \pm 0.010$ | $\mathbf{0.769} \pm 0.009$ | $\mathbf{0.933} \pm 0.011$ | $\mathbf{0.387} \pm 0.007$ |

### D.9.7 COMPUTATIONAL COMPLEXITY

We break down the computational overhead of UPCORE into the following pieces:

- Hidden State Extraction: This involves a single forward pass through the relevant layers of your pre-trained LLM for each data point in the forget set. The cost here is proportional to the size of the forget set ($|D_F|$), the dimensionality of the hidden states, and the cost of a single forward pass through the specified layers of the LLM.

Table 14: Statistical significance ($p$-values) of performance differences between UPCORE and baseline selection strategies evaluated averaged across the three unlearning methods: Gradient Ascent, Refusal, NPO on the Counterfact dataset.

| Compared Method | $p$-value |
|---|---|
| Complete | $5.51 \times 10^{-22}$ |
| Random | $3.89 \times 10^{-15}$ |
| $D^2$-Pruning | $5.04 \times 10^{-18}$ |

Table 15: Evaluation metrics from Table 2 averaged across topics in Counterfact shown for Gradient Ascent, assessed for robustness to **rephrased** and **jailbreak** variants of the forget data with the same utility data.

| Method | Selection | Retain | Neigh | Real World | Real Authors | Model Utility |
|---|---|---|---|---|---|---|
| Jailbreak | Complete | 0.417 | 0.474 | 0.599 | 0.743 | 0.291 |
| | Random | 0.430 | 0.470 | 0.629 | 0.787 | 0.305 |
| | UPCORE | **0.455** | **0.512** | **0.665** | **0.819** | **0.335** |
| Rephrase | Complete | 0.357 | 0.431 | 0.533 | 0.655 | 0.257 |
| | Random | 0.361 | 0.426 | 0.536 | 0.665 | 0.262 |
| | UPCORE | **0.376** | **0.449** | **0.555** | **0.673** | **0.279** |

- Isolation Forest Complexity: According to the original Isolation Forest paper (Liu et al., 2008b), the average training complexity is $O(T\Psi \log \Psi)$ where $T$ = number of trees, $\Psi$ = subsample size. Even for large datasets, practical complexity is close to linear in the number of samples because trees are shallow and built from small random subsamples. Isolation Forest requires only storing the trees and does not compute or store pairwise distances or full similarity matrices (unlike kNN or density-based methods). It's commonly cited as a lightweight, CPU-friendly algorithm that does not require a GPU for typical dataset sizes (e.g., scikit-learn's implementation runs efficiently on CPUs for millions of samples).

**Related works: Unlearning Methods for LLMs.** Machine unlearning methods fall into two categories: *exact unlearning*, which ensures the model is indistinguishable from one retrained without the forget data, and *approximate unlearning*, which modifies model parameters efficiently without full retraining. Due to the high cost of retraining LLMs, most approaches—including ours—use the latter. One class of methods trains models via RLHF to produce uninformative responses (e.g., "I don't know") on forget prompts (Ouyang et al., 2022; Wen et al., 2024). Yao et al. (2023) apply gradient ascent to suppress harmful outputs, substituting them with whitespace, though this causes utility loss on benign prompts. To mitigate such degradation, Chen & Yang (2023) introduce an unlearning layer effective across tasks, while Eldan & Russinovich (2023) propose a specialized architecture for removing copyrighted content. For evaluation, Maini et al. (2024) present a benchmark we adopt. Zhao et al. (2024) show that unlearning difficulty depends on the memorization of points in the forget set and their degree of entaglement with the retain set,. Unlike UPCORE, their RUM framework refines forget sets into homogeneous subsets and unlearns them sequentially to improve unlearning. Despite these advances, managing the trade-off between unlearning and utility remains challenging. We address this by introducing a data-driven coreset selection framework to minimize collateral damage.

## D.10 UPCORE FOR DIVERSE REAL-WORLD UNLEARNING

While our current setup effectively demonstrates the impact of within-topic diversity, we emphasize that UPCORE's core principle – identifying and pruning outliers based on representation variance – is broadly applicable to diverse unlearning requests, including those involving sparse and heterogeneous data. UPCORE is designed to be method-agnostic and generalizable to multi-topic settings by identifying correlated subsets within each forget request and pruning them individually. We note

Table 16: ROUGE scores and model utility across topics from the TriviaQA dataset for a fixed epoch of Gradient Ascent. UPCORE consistently achieves higher performance on data outside the forget set, with the least degradation among methods and closest performance to the base model, while still maintaining a high forget rate.

| Method | Forget | Retain | Neigh. | Real Authors | Real World | Model Utility |
|---|---|---|---|---|---|---|
| *Base Model* | *0.990* | *0.546* | *0.792* | *1.000* | *0.872* | *0.433* |
| Complete | 0.038 | 0.281 | 0.150 | 0.631 | 0.385 | 0.203 |
| Random | 0.040 | 0.287 | 0.176 | 0.646 | 0.407 | 0.224 |
| UPCORE | 0.040 | **0.323** | **0.210** | **0.660** | **0.454** | **0.251** |

Table 17: Evaluation metrics from Table 2 averaged across topics in Counterfact shown for Gradient Ascent, assessed for robustness to **jailbreak** variants of the forget data with the same utility data.

| Method | Selection | Retain | Neigh | Real World | Real Authors | Model Utility |
|---|---|---|---|---|---|---|
| Jailbreak | Complete | 0.417 | 0.474 | 0.599 | 0.743 | 0.291 |
| | Random | 0.430 | 0.470 | 0.629 | 0.787 | 0.305 |
| | RUM | 0.398 | 0.416 | 0.532 | 0.725 | 0.259 |
| | $D^2$-pruning | 0.435 | 0.479 | 0.643 | 0.792 | 0.307 |
| | UPCORE | **0.455** | **0.512** | **0.665** | **0.819** | **0.335** |

that even real-world cases such as GDPR or copyright takedown requests often involve semantically coherent topics. For example, a GDPR request may concern all information related to a single individual, which naturally forms a topic cluster. A copyright takedown may require unlearning content from a specific book or author (e.g., "Harry Potter"), again forming a semantically cohesive group. To further demonstrate generality, we perform experiments on sequential multi-topic unlearning, where we sequentially unlearn data from a multi-topic dataset formed by combining three Counterfact topics, applying UPCORE-based pruning independently to each. Each step involves a semantically coherent forget set (more realistic than isolated datapoints), while the topics across steps are unrelated, mirroring real-world requests that arrive over time and span diverse content. In this setting, Table 6 shows that UPCORE outperforms the baselines, consistently yielding higher AUC (model utility). These results confirm that UPCORE is robust to evolving, heterogeneous unlearning streams while preserving its advantages in structured topic-based forgetting, making it well-suited for real-world privacy and compliance scenarios.

**Sensitivity analysis.** We perform a sensitivity analysis of UPCORE by varying the number of Isolation Forest trees from 100 to 400. As shown in Table 18, the performance across all five metrics remains remarkably stable, with all values staying within the expected standard deviation ranges. This demonstrates that UPCORE is robust to changes in the number of trees and that its deletion and utility behavior does not depend sensitively on this hyperparameter.

### D.11 CORRELATION OF HIDDEN-STATE VARIANCE AND UTILITY.

We find that hidden state variance is strongly negatively correlated with model utility after unlearning (Pearson $r = -0.714$), supporting its use as a proxy for unlearning-induced degradation. Appendix D.9.2 shows a consistent inverse relationship between variance and AUC across topics, indicating that minimizing variance improves deletion-utility trade-offs. As shown in Fig. 7a (top left), plotting cluster variance against the utility metric from Maini et al. (2024) after fixed unlearning steps reveals a clear negative correlation.

### D.12 RESULTS ON TOFU DATASET.

Table 19 reports detailed utility metrics on the TOFU dataset under Gradient Ascent unlearning. Across all evaluation dimensions (Retain, Neighborhood, Real World, Real Authors), UPCORE consistently achieves the highest AUC scores. This confirms that variance-based pruning generalizes

Table 18: Sensitivity analysis of UPCORE across different numbers of Isolation Forest trees. Metrics are the same as those reported in Table 2.

| UPCORE Configuration | Retain | Neigh | Real World | Real Authors | Model Utility |
|---|---|---|---|---|---|
| UPCORE (100 trees) | 0.523 | 0.608 | 0.769 | 0.933 | 0.387 |
| UPCORE (150 trees) | 0.518 | 0.612 | 0.766 | 0.929 | 0.383 |
| UPCORE (200 trees) | 0.527 | 0.604 | 0.773 | 0.936 | 0.390 |
| UPCORE (300 trees) | 0.516 | 0.610 | 0.764 | 0.931 | 0.384 |
| UPCORE (400 trees) | 0.530 | 0.606 | 0.770 | 0.934 | 0.389 |

Table 19: Evaluation metrics from Table 2 across on TOFU dataset shown for Gradient Ascent.

| Method | Selection | Retain | Neigh | Real World | Real Authors | Model Utility |
|---|---|---|---|---|---|---|
| | Complete | 0.755 | 0.697 | 0.828 | 0.856 | 0.527 |
| Jailbreak | Random | 0.792 | 0.732 | 0.877 | 0.893 | 0.552 |
| | RUM | 0.671 | 0.626 | 0.744 | 0.769 | 0.469 |
| | $D^2$-pruning | 0.775 | 0.729 | 0.861 | 0.882 | 0.556 |
| | UPCORE | **0.827** | **0.758** | **0.885** | **0.899** | **0.581** |

beyond Counterfact and TriviaQA, maintaining robustness and effectiveness even when unlearning a synthetic dataset like TOFU (Maini et al., 2024).

### D.13 DISENTANGLING VARIANCE FROM OTHER FACTORS

To further disentangle the role of variance from other potential factors, we additionally run experiments where we cluster examples into groups based on multiple attributes, each measured by a concrete, factor-specific metric: 1) Answer length: total number of tokens in the ground truth answer 2)Lexical diversity: type-token ratio (TTR) computed on the ground truth answer. We find that Correlation (Answer Length vs Utility Drop) is -0.046 and Correlation (TTR vs Utility Drop) is 0.032. As the results show, we do not observe a strong correlation between either lexical diversity (TTR) or answer length and the resulting utility drop, suggesting that variance in these factors is not the primary driver of collateral damage.

Table 20: Relationship between answer length and utility drop. No strong correlation is observed, suggesting answer length is not the primary driver of collateral damage.

| Answer Length | Utility Drop |
|---|---|
| 8 | 4.8 |
| 14 | 2.1 |
| 22 | 4.2 |
| 35 | 6.2 |
| 50 | 2.1 |
| 65 | 5.5 |
| 80 | 3.0 |

Table 21: Relationship between lexical diversity (TTR) and utility drop. Variance in TTR does not strongly drive collateral damage.

| TTR | Utility Drop |
|------|------|
| 0.32 | 4.7 |
| 0.38 | 2.0 |
| 0.45 | 2.9 |
| 0.52 | 3.8 |
| 0.60 | 1.1 |
| 0.67 | 5.4 |
| 0.75 | 3.2 |

