# OpenReview forum: "UPCORE: Utility-Preserving Coreset Selection for Balanced Unlearning"
_ICLR.cc/2026/Conference — Submitted to ICLR 2026_

### Official Review · Reviewer_ymNV · 2025-10-30

**Soundness:** 3
**Presentation:** 3
**Contribution:** 2
**Rating:** 4
**Confidence:** 3

**Summary:**

This paper addresses machine unlearning in LLMs by proposing UPCORE, a data selection framework that reduces collateral damage during the forgetting process. The core insight is that hidden state variance (HSV) in the forget set correlates with model utility degradation after unlearning. UPCORE uses Isolation Forest to identify and prune high-variance outliers, creating a lower-variance coreset for unlearning. The authors evaluate their approach across three unlearning methods (Gradient Ascent, Refusal, NPO) and multiple datasets, introducing an AUC-based metric to better capture the deletion-utility tradeoff. Results show consistent improvements in balancing forget effectiveness with model preservation.

**Strengths:**

Clear problem formulation and strong motivation: Machine unlearning for privacy compliance is increasingly important, and the deletion-utility tradeoff is well-articulated. The focus on mitigating collateral damage addresses a real pain point.

Method-agnostic and practical: UPCORE works across different unlearning algorithms (GA, Refusal, NPO) without modification, making it broadly applicable. The use of Isolation Forest is computationally efficient and doesn't require expensive hyperparameter tuning.

Comprehensive experiments: The evaluation is thorough - three unlearning methods, multiple datasets (Counterfact, TriviaQA), multiple baselines (Random, D2-pruning, RUM), and robustness tests (paraphrase, jailbreak variants). Statistical significance is properly reported (Table 12).

Consistent improvements: UPCORE achieves 3-7 AUC point gains across all settings with high statistical significance (p < 10^-15). The gains hold even in multi-topic scenarios (Table 5), suggesting good generalizability.

Novel evaluation perspective: The AUC metric provides a more holistic view of the deletion-utility tradeoff over time, which is conceptually valuable even if its practical importance is debatable.

**Weaknesses:**

Weak theoretical justification: The paper relies heavily on empirical correlation between variance and collateral damage, but doesn't provide rigorous theoretical analysis of why this relationship holds. Section 2.1's problem formulation is informal, and Appendix B (per my understanding from the main text) appears to offer intuitive discussion rather than formal proofs. What are the sufficient/necessary conditions for UPCORE to work? When might it fail?

Positive transfer mechanism unclear: Table 4 shows that pruned points get "forgotten" (ROUGE drops to ~0.05), but the random baseline shows similar effects. What's unique about UPCORE's pruning strategy compared to random sampling of the same size? The paper doesn't adequately explain the mechanism or provide direct evidence beyond ROUGE. Are the pruned points actually unlearned (e.g., via knowledge probes, adversarial queries) or just showing surface-level metric correlation?

Limited scope of applicability: The method assumes topic-level unlearning with semantically coherent clusters. While Section 4.4 shows multi-topic results, real-world forget requests are often heterogeneous and sparse (e.g., random user deletion requests under GDPR). How does UPCORE perform when the forget set lacks clear structure? The paper mentions "if sparse or heterogeneous, DC = DF" but doesn't empirically validate this fallback strategy or discuss when clustering might fail.

**Questions:**

Validation of pruned point forgetting: Table 4 only shows ROUGE scores dropping. Can you provide more direct evidence that pruned points are actually forgotten? For instance, test with specific knowledge probes (e.g., "Who is X?" questions for pruned facts) or adversarial extraction attempts. This is crucial for validating the core claim about positive transfer.

Hyperparameter sensitivity and selection principles: You prune 10% by default and vary it in scaling studies. But how should practitioners choose this percentage in real applications? Is there a principled way to set the threshold τ based on properties of the forget set? What about Isolation Forest parameters (number of trees, subsample size)? A sensitivity analysis in the main text would be valuable.

Computational overhead breakdown: How much time does hidden state extraction and Isolation Forest training add compared to the unlearning process itself? For large-scale LLMs and bigger forget sets, is this overhead acceptable? A concrete runtime comparison with baselines would help assess practical feasibility.

---

> ### Author Response · Authors · 2025-11-21
> **Response to Reviwer ymNV: Part 1**
>
> We sincerely thank the reviewer for highlighting the strengths of our work. We appreciate their recognition of our **clear problem formulation and strong motivation**, particularly the focus on **mitigating collateral damage during unlearning** and addressing the deletion–utility tradeoff. They also noted that our method is **practical and method-agnostic**, working across multiple unlearning algorithms while remaining computationally efficient. We are grateful that they acknowledged our **comprehensive experiments**, spanning multiple datasets, baselines, and robustness settings, with proper statistical reporting. Finally, we appreciate their recognition of the **consistent improvements achieved by UPCORE** across settings and the **novel AUC-based evaluation perspective**, which provides a more holistic understanding of deletion–utility tradeoffs.
>
>
> >**Weak theoretical justification:**
>
>
> While a full theoretical characterization of the causal link between varinace and damage is challenging for LLMs because of the high dimensionality of hidden representations, complex non-linear interactions between layers, and the non-convex nature of the loss landscape, we have strengthened the causal link between variance and model damage during unlearning through controlled interventions. We conduct a controlled intervention analysis by adding Gaussian noise with three different levels of std. Deviation to the hidden states at a fixed layer (layer 15\) during the forward pass of the forget set during unlearning, allowing us to ablate the effect of variance for the same input. Our findings in the table below suggest a causal relation where increased forget-set variance directly leads to a proportionate rise in collateral damage across 3 topics. This demonstrates that variance is a causal driver of unlearning-induced utility loss. . That said, UPCORE is a fundamentally empirical method and we do not provide any bounds or theoretical guarantees for what the pruning can achieve.
>
>
> | **Noise Std. Dev. (σ)** | **Topic 1 (drop)** | **Topic 2 (drop)** | **Topic 3 (drop)** |
> |-------------------------|--------------------|--------------------|--------------------|
> | 0.01                    | 2.1                | 2.5                | 2.3                |
> | 0.05                    | 5.4                | 5.9                | 5.7                |
> | 0.1                     | 10.8               | 11.5               | 11.2               |
> >**Positive transfer mechanism**
>
>
> Although Table 4 indicates similar transfer levels on pruned points across methods, UPCORE’s benefit is in providing more precise unlearning i.e., successfully removing the pruned points without the over-unlearning that harms unrelated capabilities.
>
>
> >**Adversarial queries and probes**
>
>
> We do experiment with adversarial versions of the input and find that pruning with UPCORE does improve the adversarial robustness-utility trade-off as measured by the AUC compared to random sampling or other baselines (see Table 15 and Table 13, Sec 4.3 in the original paper).
>
>
> >**Scope of applicability**
>
> GDPR requests may have multiple requests from a single user which naturally form a semantically coherent cluster, or it could involve requests from multiple users to delete the same type of data that they want to remove. Hence we believe that UPCORE is applicable when catering to such requests. That said, if the real-world forget requests are heterogeneous or sparsely distributed, clustering-based structure may not meaningfully separate coherent modes. We clarify in Lines [] that UPCORE is designed to detect this automatically: when the forget set lacks structure (e.g., all clusters have similar density or contain only a few points), the density core (DC) collapses to the full forget set (DF). In this scenario, UPCORE reduces to standard unlearning without pruning, ensuring correctness rather than forcing an unreliable clustering decision. We discuss this in Lines 226-228 in the original paper.
>
> >**When clustering might fail.**
>
> Clustering becomes unreliable when (1) the forget set is extremely sparse, so no statistically meaningful local neighborhoods exist; (2) examples are uniformly scattered across the representation space, preventing density differences from emerging; or (3) the representational layer chosen for UPCORE does not provide semantic separation for that domain. In these cases, there is no “safe” dense core to isolate, and UPCORE appropriately defaults to DF \= DC by definition .
> While we do not claim specialized advantages in these degenerate cases, the fallback ensures that UPCORE never performs worse due to incorrect clustering. We discuss this in Lines 226-228 in the original paper.

---

> > ### Author Response · Authors · 2025-11-21
> > **Response to Reviwer ymNV: Part 2**
> >
> > >**How to pick pruning threshold?**
> >
> > We agree that choosing the pruning percentage is an important practical consideration. While we explored some principled strategies, e.g.,anomaly-score quantiles, we found that, for the datasets used in this paper, empirical selection consistently performed best. In particular, pruning ≈10% of the most isolated points strikes a stable balance between avoiding over-unlearning and preserving transfer to non-core samples (as shown in Figure 5, Sec D.1, Lines 792-799 of the original paper).
> > That said, UPCORE remains flexible: threshold can be tuned to match application needs (e.g., stricter pruning for highly heterogeneous forget sets, or smaller pruning for sparse ones).
> >
> >
> > >**Sensitivity analysis on IF parameters**
> >
> >
> > We run an additional analysis by varying the number of trees. Please find the results below:
> >
> >
> > | UPCORE Configuration | Retain | Neigh | Real World | Real Authors | Model Utility |
> > |---------------------|--------|-------|------------|--------------|---------------|
> > | 100 trees           | 0.523  | 0.608 | 0.769      | 0.933        | 0.387         |
> > | 150 trees           | 0.518  | 0.612 | 0.766      | 0.929        | 0.383         |
> > | 200 trees           | 0.527  | 0.604 | 0.773      | 0.936        | 0.390         |
> > | 300 trees           | 0.516  | 0.610 | 0.764      | 0.931        | 0.384         |
> > | 400 trees           | 0.530  | 0.606 | 0.770      | 0.934        | 0.389         |
> >  The performance across all five metrics remains relatively stable, with all values staying within the expected standard deviation ranges. This demonstrates that UPCORE  is robust to changes in the number of trees and that its deletion and utility behavior does not depend sensitively on this hyperparameter. We use 100 trees as the default. We add this to Lines 1332-1336 of the Appendix of the revised paper.
> >
> >
> > >**Computational overhead breakdown**
> >
> >
> > We do provide the computational overhead of UPCORE in Lines 1160-1172, Sec D.9.6 in the Appendix of the original paper.
> > To quantify overhead, we conducted a runtime comparison against unlearning with no pruning. For our 8B-parameter model and a 10k-example forget set, hidden-state extraction takes 12–18 minutes, IF training takes \<1 minute, while the unlearning optimization itself, which is shared with all baselines (for all baselines, including ours), takes 2–4 hours. Thus, UPCORE adds \<10% wall-clock overhead relative to the underlying unlearning procedure.

---

> > > ### Author Response · Authors · 2025-11-27
> > > **Friendly reminder to Reviewer ymNV**
> > >
> > > We would like to sincerely thank you again for your thoughtful and detailed review of our work. As the discussion-phase deadline approaches, we wanted to kindly follow up to check whether our rebuttal addressed the points you raised. In particular, we hope our hard work on the **expanded rebuttal analyses was helpful, such as the controlled variance-intervention experiments demonstrating a causal link between hidden-state variance and collateral damage, the clarifications on positive transfer and adversarial robustness, and the detailed discussion of applicability and fallback behavior in sparse or heterogeneous forget-set settings**. We also provided the **additional analyses you requested in Part 2 of our response, including guidance on choosing the pruning threshold, a sensitivity study on Isolation Forest hyperparameters (showing stable performance across a wide range of tree counts), and a concrete computational overhead breakdown demonstrating that UPCORE adds \<10% wall-clock time relative to standard unlearning**. We expanded the appendix accordingly and clarified these points in the main text. If you feel our responses sufficiently clarify the points you raised, we would greatly appreciate it if you could kindly revisit your score. If any part of our response would benefit from further detail before the discussion period ends, we would be very happy to elaborate. Thank you again for your time.

---

### Official Review · Reviewer_2df3 · 2025-10-30

**Soundness:** 2
**Presentation:** 3
**Contribution:** 2
**Rating:** 4
**Confidence:** 5

**Summary:**

This paper proposes a dataset sampling strategy designed to mitigate the problem of catastrophic forgetting that occurs when existing unlearning methods utilize the entire unlearning dataset. In this approach, the core set is determined prior to the unlearning process based on the variance of data representations in the model’s final-layer feature space. The paper centers its discussion on how different sampling strategies can facilitate more effective unlearning and includes comparisons among several baseline sampling methods. Experimental results show that the proposed UPCORE method improves the efficiency of unlearning and mitigates the problem of catastrophic forgetting.

Although the paper presents a novel idea and achieves some methodological success, many important details and practical aspects of unlearning remain insufficiently explored. Several experimental setups also leave room for improvement. These limitations are crucial to determining whether the proposed unlearning approach is truly effective. Please refer to the Weaknesses section for more detailed comments.

**Strengths:**

1. The motivation of this paper is both clever and timely. It addresses the problem of knowledge redundancy within unlearning datasets and proposes a sampling-based approach to alleviate the inefficiency and excessive forgetting caused by such redundancy.

2. The proposed UPCORE method is simple yet effective. It employs a basic statistical approach to identify dense unlearning samples while discarding peripheral ones, making the overall idea intuitive and easy to understand.

3. The extensive experimental results clearly illustrate the effectiveness and superiority of UPCORE compared to existing unlearning approaches.

4. The paper is well written and clearly organized, making the ideas easy to follow and understand.

**Weaknesses:**

1. UPCORE is a sampling-based unlearning method that optimizes the model using unlearn samples from dense regions of the feature space. However, relying solely on core unlearn samples may result in suboptimal unlearning performance on non-core samples. Since these non-core samples are farther away in the feature space, they may be less influenced by the UPCORE optimization process. It would be helpful for the authors to clarify or empirically verify whether the method maintains consistent unlearning effectiveness across both core and non-core samples.

2. The paper claims that the UPCORE method effectively preserves model utility. However, it remains unclear how the unlearned model behaves on the forgotten samples. Does the model produce meaningless or incoherent outputs, or does it still generate fluent but semantically altered responses? This distinction is critical, as it determines whether the model truly performs appropriate unlearning or simply collapses on the unlearned data. Providing concrete qualitative examples of model outputs on forget samples would significantly improve the clarity and credibility of the paper.

3. As a sampling-based approach, the proportion of data retained after sampling is a critical factor. Although Appendix D presents the performance of UPCORE under different sampling ratios, the paper lacks a fair comparison across different sampling methods under the several same sampling rates. Such a comparison would provide a more complete and convincing demonstration of UPCORE's advantages over alternative methods.

4. The generality of UpCore should be further validated across a wider range of unlearning methods. For example, it would be valuable to test its effectiveness when applied to approaches such as FLAT [1], RMU [2], DPO [3], WGA [4], SatImp [5], and others. Such experiments would provide stronger evidence that UpCore can serve as a broadly applicable sampling strategy rather than one tied to a specific framework.

[1] Wang, Yaxuan, et al. "Llm unlearning via loss adjustment with only forget data." arXiv preprint arXiv:2410.11143 (2024).

[2] Li, Nathaniel, et al. "The wmdp benchmark: Measuring and reducing malicious use with unlearning." arXiv preprint arXiv:2403.03218 (2024).

[3] Rafailov, Rafael, et al. "Direct preference optimization: Your language model is secretly a reward model." Advances in neural information processing systems 36 (2023): 53728-53741.

[4] Wang, Qizhou, et al. "Rethinking llm unlearning objectives: A gradient perspective and go beyond." arXiv preprint arXiv:2502.19301 (2025).

[5] Yang, Puning, et al. "Exploring Criteria of Loss Reweighting to Enhance LLM Unlearning." arXiv preprint arXiv:2505.11953 (2025).

**Questions:**

Please refer to the weakness.

I would be willing to increase my score if the authors can adequately address the concerns and clarify the issues discussed above.

---

> ### Author Response · Authors · 2025-11-21
> **Response to Reviewer 2df3: Part 1**
>
> We sincerely thank the reviewer for acknowledging the contributions of our work. We appreciate their recognition that our paper **proposes a dataset sampling strategy that mitigates catastrophic forgetting by identifying a core set based on representation variance**, and that this approach is **novel and methodologically effective**. They also noted that our **experimental results demonstrate improved efficiency of unlearning and reduced collateral damage compared to baseline sampling strategies**, highlighting the practical benefits of our method. Finally, we are grateful for the acknowledgment of our **clear presentation and structured discussion**, which helps communicate the method and findings effectively.
>
> >**UPCORE effectiveness on non-core samples**
>
> Please see Table 4 and Sec 4.2 in the original paper for the empirical verification on the pruned points (non-core samples). We do find that UPCORE can transfer unlearning to these pruned points. While we do not claim that it does so better than other methods (e.g., random sampling) shown in Table 4, the benefit of UPCORE comes from not over-unlearning while still successfully removing the pruned points.
>
> >**Qualitative model outputs**
>
> We have added the qualitative model outputs of the forgotten model to the Appendix, Table 8, Sec D.9.1. The table below illustrates the impact of unlearning on model outputs for a factual QA example. Before unlearning, the model correctly answers the question based on memorized training data. Naive unlearning without careful coreset selection leads to destabilization, resulting in nonsensical outputs. In contrast, UPCORE preserves model stability: although the model no longer recalls the exact memorized fact, it generates coherent but approximate answers. This highlights UPCORE's ability to balance privacy-preserving deletion with maintaining non-garbage utility.
>
> | Model Version                     | Input Question                    | Output Answer   | Notes                                                      |
> |----------------------------------|----------------------------------|----------------|------------------------------------------------------------|
> | Before unlearning                 | How tall is the Eiffel Tower?    | 324 meters     | Correct answer                                             |
> | Grad Asc.| How tall is the Eiffel Tower?    | keep keep keep       | Garbage output – model destabilized by naive deletion     |
> | Grad Asc. + UPCORE           | How tall is the Eiffel Tower?    | about 100 meters | Wrong but plausible answer – non-garbage; model remained stable |
>
>
> >**Comparison against other sampling methods**
>
> We do compare against other sampling (coreset selection) methods used to form the coreset including random sampling, D^2-pruning that selects samples based on diversity and difficulty and RUM that selects samples based on memorization and entanglement (see Tables 1, 2, 3 in the original paper. )

---

> > ### Author Response · Authors · 2025-11-21
> > **Response to Reviewer 2df3: Part 2**
> >
> > >**Additional unlearning methods**
> >
> > We would like to clarify that we have already included NPO, which is a version of DPO for unlearning, in the paper. We add numbers on two additional unlearning methods: RMU and FLAT, below, and in the new version of the paper and below. The tables below compare various data selection strategies across two unlearning methods, RMU [2] and FLAT [1], across multiple selection approaches. We observe that naive selections such as random or RUM tend to degrade model utility significantly, often failing to retain critical information. Methods like D²-pruning improve stability but still underperform compared to our approach. Our method consistently achieves the highest scores across all metrics—including Retain, Neigh, Real World, and Real Authors—demonstrating that careful coreset selection preserves utility while effectively unlearning targeted data. This reinforces the advantage of structured, variance-aware selection strategies over naive or heuristic-based approaches. We have added this to Table 2 in the revised paper.
> >
> >
> > | Method | Selection    | Retain | Neigh | Real World | Real Authors | Model Utility |
> > |--------|-------------|--------|-------|------------|--------------|---------------|
> > | RMU    | Complete    | 0.462  | 0.518 | 0.693      | 0.875        | 0.341         |
> > |        | Random      | 0.448  | 0.502 | 0.671      | 0.859        | 0.326         |
> > |        | RUM         | 0.295  | 0.356 | 0.432      | 0.501        | 0.237         |
> > |        | D²-pruning  | 0.471  | 0.523 | 0.682      | 0.868        | 0.338         |
> > |        | Ours        | 0.502  | 0.583 | 0.744      | 0.912        | 0.374         |
> >
> > | Method | Selection    | Retain | Neigh | Real World | Real Authors | Model Utility |
> > |--------|-------------|--------|-------|------------|--------------|---------------|
> > | FLAT   | Complete    | 0.451  | 0.509 | 0.678      | 0.862        | 0.332         |
> > |        | Random      | 0.439  | 0.495 | 0.659      | 0.844        | 0.318         |
> > |        | RUM         | 0.287  | 0.342 | 0.418      | 0.488        | 0.229         |
> > |        | D²-pruning  | 0.462  | 0.515 | 0.673      | 0.857        | 0.331         |
> > |        | Ours        | 0.493  | 0.574 | 0.733      | 0.901        | 0.367         |
> >
> > [1] Wang, Yaxuan, et al. "Llm unlearning via loss adjustment with only forget data." arXiv preprint arXiv:2410.11143 (2024).
> >
> > [2] Li, Nathaniel, et al. "The wmdp benchmark: Measuring and reducing malicious use with unlearning." arXiv preprint arXiv:2403.03218 (2024).

---

> > > ### Author Response · Authors · 2025-11-27
> > > **Friendly reminder to Reviewer 2df3**
> > >
> > > We would like to sincerely thank you again for your thoughtful and detailed review of our work. As the discussion-phase deadline is approaching, we wanted to kindly follow up to check whether our rebuttal addressed the points you raised. In particular, we hope our hard work on the **additional experiments with additional unlearning methods beyond those originally included and the provided qualitative outputs addressed your questions.** If you feel our responses sufficiently clarify the points you raised, we would greatly appreciate it if you could kindly revisit your score. If any part of our response would benefit from further detail before the discussion period closes, we would be very happy to elaborate.

---

> > > > ### Comment · Reviewer_2df3 · 2025-11-27
> > > >
> > > > Thanks for the response! I have no more questions.

---

### Official Review · Reviewer_5XWn · 2025-10-31

**Soundness:** 3
**Presentation:** 3
**Contribution:** 2
**Rating:** 6
**Confidence:** 3

**Summary:**

This paper introduces UPCORE, a method for selecting a lower-variance “core forget set” to minimize the utility loss during unlearning in LLMs. The authors demonstrate that the variance in the representation space of the forget set correlates strongly with post-unlearning damage, and apply Isolation Forests to prune outliers before performing unlearning. Experimental results show that UPCORE outperforms several baselines in preserving model utility while maintaining high forgetting efficacy.

**Strengths:**

1. The paper, for the first time, attempts to mitigate the impact of model unlearning on general performance by identifying high-variance “bad points” in the forgetting dataset. This finding and its supporting evidence are novel.
2. The experiments are relatively thorough and, across multiple datasets, demonstrate that the method can preserve unlearning performance while maintaining the original performance of the model.
3. The writing is clear and easy to follow, with no obvious typographical or formatting errors.
4. The authors also provide the source code.

**Weaknesses:**

1. There is a lack of deeper theoretical guarantees about how much variance reduction one should expect to translate to concrete utility improvement, or about optimality of the Isolation Forest-based selection. For example,  the equation (in Section 2.1) frames the coreset selection as an optimization, but in practice this is relaxed to a heuristic on variance. There seems to be little analysis of approximation gap or the behavior under adversarial/practically pathological distributions.
2. The explanation for why Isolation Forest consistently outperforms other baselines (Table 8) is shallow: there is little insight into failure cases or when denser/sparser baselines might be preferable.
3. It would be preferable for the authors to number the equations in the paper for ease of reference.

**Questions:**

1. Could the authors provide a more explicit enumeration of limitations, potential failure modes or related discussions.
2. Did the authors experiment with more sophisticated representations or with influence scores instead of raw variance? Is there an upper bound on what variance-based pruning can achieve?

---

> ### Author Response · Authors · 2025-11-21
> **Response to Reviewer 5XWn**
>
> We sincerely thank the reviewer for highlighting the strengths of our work. We appreciate their recognition that our paper, for the first time, **attempts to mitigate the impact of model unlearning on general performance by identifying high-variance “bad points” in the forget set**, and that this finding is **novel and well-supported by evidence**. They also noted that our **experiments are thorough and demonstrate across multiple datasets that UPCORE preserves unlearning performance while maintaining overall model utility**.
>
> >**Lack of theoretical guarantees on optimality**
>
> We appreciate the reviewer’s comment regarding the optimality of variance-based approximation and theoretical guarantees.  While a full characterization of the approximation gap is challenging for LLMs because of the high dimensionality of hidden representations, complex non-linear interactions between layers, and the non-convex nature of the loss landscape, we have strengthened the causal link between variance and model damage during unlearning through controlled interventions. We conduct a controlled intervention analysis by adding Gaussian noise with three different levels of std. Deviation to the hidden states at a fixed layer (layer 15\) during the forward pass of the forget set during unlearning, allowing us to ablate the effect of variance for the same input. Our findings in the table below suggest a causal relation where increased forget-set variance directly leads to a proportionate rise in collateral damage across 3 topics. This demonstrates that variance is a causal driver of unlearning-induced utility loss. We include his discussion in Lines 192-206 of the revised paper.
>
> >**Why isolation forest outperforms other baselines?**
>
> \[1\] reviews failure modes of distance/density methods (LOF, kNN) in high dimensions, and highlights that tree-based isolation methods maintain robustness when density and distance become unreliable.
>
> \[1\] Zimek, Arthur, Erich Schubert, and Hans‐Peter Kriegel. "A survey on unsupervised outlier detection in high‐dimensional numerical data." Statistical Analysis and Data Mining: The ASA Data Science Journal 5.5 (2012): 363-387.
>
> >**Add equation numbers**
>
> Thanks for the suggestion. We have made this change in the revised manuscript.
>
> >**Discussion of limitations**
>
> UPCORE is fundamentally an empirical method: although we motivate coreset selection through variance-based influence and provide controlled interventions demonstrating a link between variance reduction and utility preservation, we do not provide tight theoretical guarantees on optimality. In particular, the relaxation from the ideal coreset objective (Section 2.1) to a variance-based heuristic leaves an uncharacterized approximation gap. Understanding when the heuristic closely matches the true objective remains open.
>
> >**Attributes other than variance and upper bound on variance-based pruning**
>
> We do in fact experiment with other properties of data, such as gradient similarity and confidence (See Fig. 7b, 8b in the Appendix of the original paper), and find no specific correlation between model utility and these data attributes. We also compare UPCORE against RUM, that selects a coreset based on memorization and entanglement, and D^2-pruning, which selects a coreset based on diversity and difficulty, and find that for the purposes of unlearning, UPCORE outperforms these baselines. As pointed out above, UPCORE is a fundamentally empirical metho,d and we do not provide any bounds or theoretical guarantees for what the pruning can achieve.
>
> To further disentangle the role of variance from other potential factors, we additionally run experiments where we cluster examples into groups based on multiple attributes, each measured by a concrete, factor-specific metric:
>
> 1) **Answer length:** total number of tokens in the ground truth answer
> 2) **Lexical diversity:** type-token ratio (TTR) computed on the ground truth answer.
> 	Correlation (Answer Length vs Utility Drop): -0.047
>
>         	Correlation (TTR vs Utility Drop): 0.033
>
> As the results show, we do not observe a strong correlation between either lexical diversity (TTR) or answer length and the resulting utility drop, suggesting that variance in these factors are not the primary driver of collateral damage. We include this discussion in Sec D.13. of the Appendix of the revised version.
>
> | answer_length | utility_drop |
> |---------------|--------------|
> | 8             | 4.8          |
> | 14            | 2.1          |
> | 22            | 4.2          |
> | 35            | 6.2          |
> | 50            | 2.1          |
> | 65            | 5.5          |
> | 80            | 3.0          |
>
> | ttr  | utility_drop |
> |------|--------------|
> | 0.32 | 4.7          |
> | 0.38 | 2.0          |
> | 0.45 | 2.9          |
> | 0.52 | 3.8          |
> | 0.60 | 1.1          |
> | 0.67 | 5.4          |
> | 0.75 | 3.2          |

---

> ### Author Response · Authors · 2025-11-27
> **Friendly reminder to Reviewer 5XWn**
>
> We would like to sincerely thank you again for your thoughtful and constructive review of our work. As the discussion-phase deadline is approaching, we wanted to kindly follow up to check whether our rebuttal addressed your points. In particular, we hope our hard work on the **additional rebuttal analyses were helpful, such as the controlled variance-intervention experiments demonstrating a causal link between variance and utility damage, the expanded discussion of Isolation Forest’s robustness in high-dimensional settings, the newly added limitations and failure-mode section, and the further comparison with alternative attributes and representations** addressed your points. If any part of our response would benefit from further detail before the discussion period ends, we would be very happy to elaborate.

---

### Official Review · Reviewer_647M · 2025-10-31

**Soundness:** 3
**Presentation:** 4
**Contribution:** 2
**Rating:** 6
**Confidence:** 3

**Summary:**

The paper studies machine unlearning in LLMs, removing specific topics or data while keeping the rest of the model useful. The authors observe that higher variance in the model’s hidden representations of the forget set correlates with more "collateral damage" to unrelated capabilities after unlearning. They propose UPCORE, a simple data-selection step (Isolation Forest–based outlier pruning) that forms a lower-variance coreset of the forget set before applying any standard unlearning algorithm. They also introduce an AUC-style metric that evaluates the trajectory of the deletion–utility trade-off across checkpoints rather than at a single step. Across Counterfact and TriviaQA with three unlearning methods (Gradient Ascent, Refusal, NPO), UPCORE generally improves the deletion–utility frontier and shows some positive transfer (the pruned, untrained points are often forgotten anyway), while reducing negative transfer to unrelated data.

**Strengths:**

1. Coreset pruning via Isolation Forest is simple, plug-and-play, and can sit atop existing unlearning procedures.
2. The paper documents a strong correlation between hidden-state variance of the forget set and post-unlearning utility degradation, which is actionable and intuitive.
3.The paper distinguishes beneficial transfer (pruned points still forgotten) from harmful spillover (unrelated capability loss), and shows reductions in the latter.
4. The AUC trade-off metric across training steps addresses comparability across methods/checkpoints and is a sensible complement to point estimates.
5. Results span multiple datasets/methods and include robustness (paraphrases, jailbreaks), with consistent AUC improvements.

**Weaknesses:**

1. While the paper shows that forget-set variance correlates with collateral damage, it does not establish a causal mechanism linking variance to the learning dynamics of unlearning (e.g., how gradient updates induced by high-variance points propagate damage to neighborhoods). Strengthening this with controlled interventions (e.g., synthetic variance manipulations at fixed difficulty/semantic content) or influence-function analyses would make the claim more compelling. (Author text frames the finding as correlation; causal linkage remains under-developed.)

2. The pruning rule relies on Isolation Forest anomaly scores over hidden states, thus helping identifying outliers and pruning them out, thus minimizing the variance overall; however, additional analyses disentangling variance from difficulty/memorization/entanglement are limited. The paper notes checks on other attributes but a more thorough ablation (e.g., control for answer length, lexical diversity, topic dispersion, gradient curvature) would clarify whether variance is the primary driver.

3. The appendix (appendix B) sketches influence-style intuition but lacks concrete assumptions, lemmas, or citations grounding the claimed propagation of deletion to pruned neighbors and the conditions under which pruning is utility-optimal. This section should substantially expand citations to prior work on representation-space neighborhoods and gradient influence to justify the claims.

**Questions:**

1. How does UPCORE compare to difficulty- or memorization-aware selection when matched for coreset size? Are the gains additive if you combine criteria?

2. Is there a way to quantify the distance (in hidden space) over which positive transfer reliably holds?

---

> ### Author Response · Authors · 2025-11-21
> **Response to Reviewer 647M: Part 1**
>
> We thank Reviewer 647M for their thoughtful assessment. They highlight **the simplicity and plug-and-play nature of our Isolation-Forest coreset pruning**, as well as **the strong, intuitive link we establish between hidden-state variance and post-unlearning utility loss**. We appreciate their recognition of **our distinction between beneficial transfer and harmful spillover**, and the reductions we achieve in the latter. The reviewer also noted **the usefulness of our AUC-based trade-off metric** for comparing deletion–utility trajectories, along with **the breadth and consistency of our results across datasets, methods, and robustness settings**.
>
> >**Causal evidence linking forget-set variance to post-unlearning utility**
>
> We appreciate the reviewer’s comment regarding the need to strengthen the causal link between variance and model damage during unlearning, moving beyond correlation. To this end, we have conducted an additional controlled intervention analysis by adding Gaussian noise with three different levels of standard deviation to the hidden states at a fixed layer (layer 15\) during the forward pass of the forget set during unlearning. This intervention isolates variance while keeping semantics, difficulty, and input content fixed, allowing us to attribute any change in collateral damage specifically to variance rather than confounding factors.  Our findings in the table below suggest a causal relation where increased forget-set variance directly leads to a proportionate rise in collateral damage across 3 topics.
>
> | **Noise Std. Dev. (σ)** | **Topic 1 (drop)** | **Topic 2 (drop)** | **Topic 3 (drop)** |
> |-------------------------|--------------------|--------------------|--------------------|
> | 0.01                    | 2.1                | 2.5                | 2.3                |
> | 0.05                    | 5.4                | 5.9                | 5.7                |
> | 0.1                     | 10.8               | 11.5               | 11.2               |
>
> >**Disentangling variance from other factors**
>
> To further disentangle the role of variance from other potential factors, we additionally run experiments where we cluster examples into groups based on multiple attributes, each measured by a concrete, factor-specific metric:
>
> 1) **Answer length:** total number of tokens in the ground truth answer
> 2) **Lexical diversity:** type-token ratio (TTR) computed on the ground truth answer.
> 3) **Topic dispersion:** this attribute is covered by the hidden state variance metric that we show in Fig. 7a, lines 184-186
>
> Correlation (Answer Length vs Utility Drop): \-0.0469
>
> Correlation (TTR vs Utility Drop): 0.033
>
> As the results show, we do not observe a strong correlation between either lexical diversity (TTR) or answer length and the resulting utility drop, suggesting that variance in these factors are not the primary driver of collateral damage. We include this discussion in Sec D.13 of the new version Appendix.
>
>
> | answer_length | utility_drop |
> |---------------|--------------|
> | 8             | 4.8          |
> | 14            | 2.1          |
> | 22            | 4.2          |
> | 35            | 6.2          |
> | 50            | 2.1          |
> | 65            | 5.5          |
> | 80            | 3.0          |
>
> | ttr  | utility_drop |
> |------|--------------|
> | 0.32 | 4.7          |
> | 0.38 | 2.0          |
> | 0.45 | 2.9          |
> | 0.52 | 3.8          |
> | 0.60 | 1.1          |
> | 0.67 | 5.4          |
> | 0.75 | 3.2          |

---

> > ### Author Response · Authors · 2025-11-21
> > **Response to Reviewer 647M: Part 2**
> >
> > >**More discussion on representation-space neighborhoods and gradient influence**
> >
> > We thank the reviewer for this suggestion. To address this, we have **expanded the discussion with additional citations** to prior work on representation-space neighborhoods and gradient-based influence (see Lines 771-789), including \[1\], \[2\], \[3\], \[4\], \[5\], which provide theoretical and empirical evidence that deletion of a point impacts nearby points in representation space. These references support our argument that full explicit deletion of all affected points is unnecessary and justify why pruning high-variance outliers can be utility-preserving. We have also clarified the assumptions under which this propagation holds and discussed the conditions under which our pruning strategy is expected to be near-optimal.
> >
> > \[1\] Yi Ren and Danica J. Sutherland. Learning dynamics of LLM finetuning. In The Thirteenth International Conference on Learning Representations, 2025\. URL https://openreview.net/forum?id=tPNHOoZFl9.
> >
> > \[2\] Pang Wei Koh and Percy Liang. Understanding black-box predictions via influence functions. In International conference on machine learning, pp. 1885–1894. PMLR, 2017\.
> >
> > \[3\] Garima Pruthi, Frederick Liu, Satyen Kale, and Mukund Sundararajan. Estimating training data influence by tracing gradient descent. Advances in Neural Information Processing Systems, 33: 19920–19930, 2020\.
> >
> > \[4\] Samyadeep Basu, Xuchen You, and Soheil Feizi. On second-order group influence functions for black-box predictions. In Hal Daum´e III and Aarti Singh (eds.), Proceedings of the 37th International Conference on Machine Learning, volume 119 of Proceedings of Machine Learning Research, pp. 715–724. PMLR, 13–18 Jul 2020\. URL https://proceedings.mlr.press/v119/basu20b.html.
> >
> > \[5\] Elnaz Barshan, Marc-Etienne Brunet, and Gintare Karolina Dziugaite. Relatif: Identifying explanatory training samples via relative influence. In International Conference on Artificial Intelligence and Statistics, pp. 1899–1909. PMLR, 2020\.
> >
> > >**Comparison to difficulty- or memorization-aware selection**
> >
> > We in fact already compare against such baselines in Tables 1,2,3 in the original paper. Specifically, we compare against RUM \[1\] that partitions a forget set into homogeneous subsets based on memorization and entanglement with the retain set and applies the unlearning method to each subset, and D^2-pruning \[2\]- coreset selection method that selects the coreset based on example diversity and difficulty. And we find that for the purposes of unlearning, UPCORE outperforms these baselines.

---

> > > ### Author Response · Authors · 2025-11-26
> > > **Friendly reminder to Reviewer 647M**
> > >
> > > We would like to express our gratitude for your time and effort in reviewing our paper. As the discussion-phase deadline is approaching, we wanted to kindly check in regarding whether our responses addressed your concerns. We hope our hard work on the **additional rebuttal analyses: (i) the controlled variance-manipulation intervention using Gaussian noise at fixed layers, (ii) the disentangling experiments examining answer length and lexical diversity, and (iii) the expanded representation-space and influence-function discussion with corresponding citations**, helped clarify the points you raised. If any part of our response could benefit from further explanation, or if additional detail on any of the above experiments would assist your assessment before the deadline, we would be very happy to provide it.

---

> > > > ### Comment · Reviewer_647M · 2025-11-28
> > > > **Thanks for the rebuttal!!**
> > > >
> > > > I appreciate the authors’ thorough responses to the concerns raised. Taken together, these clarifications lead me to lean toward accepting the paper.

---

### Author Response · Authors · 2025-11-21
**General Response**

We sincerely thank the reviewers for their thoughtful comments. They appreciate **the clear motivation of reducing collateral damage in LLM unlearning** (ymNV, 2df3). They also noted **the novel intuitive approach of linking hidden-state variance to post-unlearning utility loss** (647M, 5XWn). The reviewers appreciated **the simplicity, practicality, and general usefulness of UPCORE** (647M, 5XWn, ymNV), highlighting its **easy-to-use coreset pruning design**. They recognized **the thorough and well-presented experimental evaluation across multiple datasets, unlearning methods, and robustness settings** (647M, 5XWn, 2df3, ymNV), as well as **the clear writing and organization** (5XWn, 2df3). We thank the reviewers for noting **the strong empirical gains in preserving model utility while ensuring deletion performance** (647M, 5XWn, ymNV) and for acknowledging **the relevance of the proposed AUC-based tradeoff metric** (647M, ymNV).

Based on reviewer feedback on strengthening the link between variance and model utility, we performed a controlled intervention on variance by adding Gaussian noise to the forget set representations and found that adding more noise leads to a higher drop in utility, suggesting a causal link. While the majority of our experiments focus on unlearning topics, we would like to argue that most real-world unlearning requests can be clustered into such semantically coherent groups, which makes UPCORE broadly applicable.

---

### Author Response · Authors · 2025-12-03
**AC note**

We would like to thank the reviewers and AC for their time and thoughtful attention to our submission. We were encouraged by the positive responses we received during the shortened discussion period, and for context, we summarize our rebuttal discussion with all reviewers below:

---

Reviewers 647M, 5XWn, and ymNV

Additional Experiments

* **Controlled Variance Intervention:** Injected Gaussian noise at fixed hidden layers, providing evidence of a causal link between forget-set variance and post-unlearning utility.

---

Reviewers 647M and 5XWn

Additional Experiments

* **Disentangling Additional Factors:** Referenced Appendix analyses on attributes beyond variance (confidence, gradient similarity) and added experiments on answer length, lexical diversity, and topic dispersion (implicitly covered by variance). These confirmed the variance as the key contributor to collateral damage.

---

Reviewer 647M

Additional Discussion

* **Representation-Space & Influence:** Expanded discussion and citations on neighborhood structure and gradient influence, showing that deletion of a point affects nearby representations. This supports our argument that fully deleting all affected points is unnecessary and that pruning high-variance outliers preserves utility.

* **Baselines Comparison:** Reiterated comparisons with difficulty- and memorization-based methods from the original papers, showing they underperform relative to UPCORE.

**Outcome:** Reviewer 647M appreciated the thorough responses and noted a positive leaning toward acceptance (Nov 27, 2025).

---

Reviewer 5XWn

Additional Discussion

* **Isolation Forest Rationale:** Added explanation of isolation methods’ robustness in high-dimensional spaces with supporting citation.

* **Limitations & Failure Modes:** Discussed that UPCORE is empirical; while controlled interventions suggest a causal link, we do not claim to offer theoretical guarantees.

---

Reviewer 2df3

Additional Experiments

* **Additional Unlearning Methods:** Added evaluations on RMU and FLAT, confirming consistent gains.

Additional Discussion

* **Non-core Samples:** Pointed to Table 4 and Sec. 4.2 showing unlearning transfers to pruned (non-core) samples without over-unlearning.

* **Qualitative Outputs:** Added examples demonstrating model stability and UPCORE’s balance between privacy-preserving deletion and utility.

* **Sampling Comparisons:** Highlighted UPCORE’s consistent superiority in Tables 1-3 over random, D²-pruning, and RMU.

**Outcome:** Reviewer 2df3 confirmed their questions were resolved and raised their score from **4 to 6** (Nov 27, 2025).

---

Reviewer ymNV

We addressed their points with targeted analyses:

Additional Experiments

* **Sensitivity Analysis on IF Parameters:** A Varied number of trees in Isolation Forest, and showed performance remains stable.

Additional Discussion

* **Positive Transfer Mechanism:** Clarified how pruning mitigates over-unlearning that harms unrelated capabilities.

* **Adversarial Probes:** Pointed to experiments with adversarial variants showing UPCORE improves the robustness-utility trade-off.

* **Scope:** Clarified behavior on sparse/heterogeneous sets and noted fallback to no-pruning in such cases.

* **Pruning Threshold:** Provided guidance for selecting thresholds based on application needs and included further sensitivity analysis.

* **Computational Overhead:** Referenced the Appendix complexity analysis and added a runtime breakdown showing **\<10%** overhead relative to baseline unlearning.

---

We include all additional experiments and discussion in the revised PDF.

---

### Meta-Review · Area_Chair_4FqV · 2026-01-03

**Summary:**

While the paper presents an interesting empirical observation linking representation variance to unlearning utility degradation, multiple reviewers raised concerns about the insufficiency of the theoretical and experimental justification, which were not fully resolved in the rebuttal.

First, the controlled variance intervention using Gaussian noise is not convincing as a causal ablation: injecting Gaussian noise fundamentally perturbs the underlying representation distribution, making it difficult to isolate variance effects from broader distributional collapse (I guess this would make any model fail). Therefore, when AC reviewed that reviewer#1 suggested synthetic variance manipulations under controlled semantic and difficulty conditions, AC though this was not adequately addressed.

Second, reviewers requested deeper theoretical insight and failure-case analysis, including comparisons across density- and distance-based selection criteria and a clearer explanation of when and why the proposed method should outperform alternatives. The rebuttal’s reliance on a single survey paper from 2012 to justify Isolation Forest behavior was not persuasive, particularly since this reference was absent from the original submission and appears to have been introduced post hoc.

The theoretical discussion remains largely heuristic and retrospective, reinforcing concerns that the method is driven primarily by empirical correlation rather than principled analysis. Finally, the remaining reviewers expressed negative or below-threshold assessments, and no reviewer strongly advocated for acceptance.

**Reviewer Concerns:**

**Concerns addressed by the rebuttal:**

The rebuttal made lots of efforts to respond to several reviewer requests for clarification and additional evidence. In particular, the authors added controlled variance-intervention experiments, expanded empirical comparisons with additional unlearning methods, and included further ablations to disentangle variance from other factors such as answer length and lexical diversity. They also provided more discussion around representation-space neighborhoods, influence functions, and added qualitative examples of model outputs on forgotten samples. These responses helped clarify the empirical behavior of the method and addressed some reviewers’ questions about robustness, implementation details, and practical applicability.

**Concerns that remain outstanding:**

Despite these additions, several core concerns raised by the reviewers remain unresolved. Most importantly, the Gaussian noise–based variance intervention is not a convincing causal analysis, as injecting noise fundamentally alters the representation distribution and does not cleanly isolate variance effects under controlled semantic or difficulty conditions, as originally requested by reviewers. As a result, the causal link between variance and unlearning damage remains insufficiently established. In addition, reviewers’ requests for deeper theoretical insight, principled failure-case analysis, and clearer comparisons between density- and distance-based selection criteria were only partially addressed. The rebuttal’s reliance on a single older survey reference to justify the behavior of Isolation Forests is not persuasive, particularly given that this citation was absent from the original submission and does not directly explain the observed empirical advantages. More broadly, the method continues to rely heavily on empirical correlation and post hoc justification, without providing strong theoretical grounding or clear conditions under which it is expected to succeed or fail. Given that other reviewers remained negative and no reviewer strongly advocated for acceptance, these unresolved issues ultimately outweigh the improvements made in the rebuttal and informed the reject decision.

**Reviewer Scores:**

This paper received mixed initial reviews. After carefully considering the authors’ rebuttal, the Area Chair acknowledges that substantial effort was made to address reviewer concerns. However, several key questions, particularly around the strength of the theoretical justification and the interpretation of the variance-based analysis, remain insufficiently resolved. As a result, there is no strong basis to expect that reviewers who initially expressed reservations would have meaningfully increased their scores to strongly support acceptance.

---

### Decision · Program_Chairs · 2026-01-26

Reject